# Inhibitory G proteins play multiple roles to polarize sensory hair cell morphogenesis

Amandine Jarysta[1], Abigail LD Tadenev[1], Matthew Day[1], Barry Krawchuk[1], Benjamin E Low[1], Michael V Wiles[1], Basile Tarchini[1,2]*

[1]The Jackson Laboratory, Bar Harbor, United States; [2]Tufts University School of Medicine, Boston, United States

**Abstract** Inhibitory G alpha (GNAI or Gαi) proteins are critical for the polarized morphogenesis of sensory hair cells and for hearing. The extent and nature of their actual contributions remains unclear, however, as previous studies did not investigate all GNAI proteins and included non-physiological approaches. Pertussis toxin can downregulate functionally redundant GNAI1, GNAI2, GNAI3, and GNAO proteins, but may also induce unrelated defects. Here, we directly and systematically determine the role(s) of each individual GNAI protein in mouse auditory hair cells. GNAI2 and GNAI3 are similarly polarized at the hair cell apex with their binding partner G protein signaling modulator 2 (GPSM2), whereas GNAI1 and GNAO are not detected. In *Gnai3* mutants, GNAI2 progressively fails to fully occupy the sub-cellular compartments where GNAI3 is missing. In contrast, GNAI3 can fully compensate for the loss of GNAI2 and is essential for hair bundle morphogenesis and auditory function. Simultaneous inactivation of *Gnai2* and *Gnai3* recapitulates for the first time two distinct types of defects only observed so far with pertussis toxin: (1) a delay or failure of the basal body to migrate off-center in prospective hair cells, and (2) a reversal in the orientation of some hair cell types. We conclude that GNAI proteins are critical for hair cells to break planar symmetry and to orient properly before GNAI2/3 regulate hair bundle morphogenesis with GPSM2.

*For correspondence:
basile.tarchini@jax.org

Competing interest: The authors declare that no competing interests exist.

## eLife assessment

This study examines an **important** aspect of the development of the auditory system, the role of guanine nucleotide-binding protein subunits, GNAIs, in stereociliary bundle formation and orientation, by examining bundle phenotypes in multiple compound GNAI mutants. The experiments are highly rigorous and thorough and include detailed quantifications of bundle morphologies and changes. The depth and care of the study are impressive, with **convincing** results regarding the roles of GNAIs in stereociliary bundle development. Further, the reviewers believe this to be the definitive study of the role of GNAIs in bundle orientation and development.

## Introduction

Developing sensory hair cells (HC) in the inner ear undergo a complex polarization process to detect and interpret mechanical stimuli, including sound. Each mature HC is able to detect stimuli in a directional manner by developing an asymmetric brush of actin-based membrane protrusions, or stereocilia: the hair bundle. Neighboring HC also adopt concerted orientations to align their hair bundles, a property known as planar cell polarity (PCP). One subclass of heterotrimeric guanine nucleotide-binding (G) protein was intimately associated with different levels of mouse HC polarization: inhibitory G alpha subunits (GNAI1, GNAI2, GNAI3, and GNAO; collectively GNAI or Gαi) (*Figure 1A*).

**Figure 1.** Summary of GNAI-related functions proposed previously in hair cells (HC). (**A**) Phylogenetic tree of GNAI/O proteins with percent amino acid identity (mouse). (**B**) Apical HC differentiation from symmetry breaking to hair bundle development. The distribution of the GPSM2-GNAI complex at the bare zone and stereocilia tips is indicated in orange. Arrows indicate off-center (left) and then inward (middle) movements of the basal body. (**C**) Defects observed with pertussis toxin (ptx) or when inactivating GNAI proteins. Defective off-center migration of the basal body and inverted OHC1–2 were only observed with ptx, respectively, in cochlear explants (in vitro) and by expressing the ptx catalytic subunit (ptxA) in vivo. Mouse knock-out (KO)s of *Gnai* genes were to date only reported to affect hair bundle morphogenesis. Known GNAI regulators that produce similar defects when inactivated are indicated on top for each type of defect. DKO, double KO.

However, several roles proposed for GNAI proteins have not been validated physiologically, and the individual contribution of each GNAI remains uncertain.

HC polarization along the epithelial plane starts with the off-center migration of the basal body and its associated primary cilium, the kinocilium (***Denman-Johnson and Forge, 1999***; ***Mbiene and Sans, 1986***; ***Tilney et al., 1992***). At that stage, regulator of G protein signaling 12 (RGS12) is required for GNAI and the G protein signaling modulator 2 (GPSM2) scaffold to form a polarized complex at the apical membrane on the side of the off-center basal body (***Akturk et al., 2022***; ***Ezan et al., 2013***; ***Tarchini et al., 2013***). GPSM2-GNAI is best known as a highly conserved protein complex orienting the mitotic spindle during progenitor divisions (***Du et al., 2001***; ***Schaefer et al., 2001***; ***Schaefer et al., 2000***; ***Woodard et al., 2010***). In post-mitotic HC, GPSM2-GNAI first excludes microvilli and microvilli-derived stereocilia from the portion of the HC surface where it resides, the expanding bare zone (***Figure 1B***). Bare zone expansion then pushes back the basal body/kinocilium from a more eccentric position near the lateral junction to a less eccentric position at the vertex of the forming hair bundle. Later, GPSM2-GNAI becomes enriched at the distal tip of row 1 stereocilia abutting the

bare zone (*Tarchini et al., 2016*). In these stereocilia, GPSM2-GNAI is a module of the elongation complex that also comprises MYO15A, WHRN, and EPS8 (*Mauriac et al., 2017*; *Tadenev et al., 2019*). GPSM2-GNAI is required at row 1 tips for boosting enrichment of other elongation complex partners, and presumably actin incorporation, compared to further stereocilia rows. GPSM2-GNAI thus confers row 1 its tallest identity and the hair bundle its asymmetric graded-height morphology.

Pertussis toxin (ptx) has been extensively used as a tool to ADP-ribosylate the GNAI subunit and dissociate heterotrimeric Gαiβγ protein complexes from G protein-coupled receptors (GPCR) to inactivate downstream signaling (*Locht et al., 2011*). In vivo expression of ptx catalytic subunit (ptx-S1 or ptxA) prevents normal enrichment and polarization of GPSM2-GNAI in developing HC (*Tadenev et al., 2019*; *Tarchini et al., 2013*), suggesting that ADP-ribosylation directly or indirectly inhibits GPSM2-GNAI function as well. Ptx provokes immature-looking hair bundles with severely stunted stereocilia, mimicking defects in *Gpsm2* mutants and *Gnai2*; *Gnai3* double mutants (*Beer-Hammer et al., 2018*; *Mauriac et al., 2017*; *Tadenev et al., 2019*; *Tarchini et al., 2016*). In contrast, stereocilia height is more variably reduced in *Gnai3* single mutants, with defects more severe at the cochlear base (*Beer-Hammer et al., 2018*; *Mauriac et al., 2017*). This can explain why hearing loss is more severe at high frequencies in *Gnai3* mutants, but profound at all frequencies in a *ptxA* model and in mutants lacking GPSM2 or both GNAI2 and GNAI3 (*Beer-Hammer et al., 2018*; *Mauriac et al., 2017*; *Tarchini et al., 2016*). Surprisingly, before affecting hair bundle differentiation at postnatal stages, ptx also causes two distinct defects in HC polarization at embryonic stages.

First, one study reported that a high dose of ptx in cultured explants of the developing cochlea results in a low proportion of symmetrical HC with a central kinocilium surrounded by a rounded hair bundle (*Figure 1C*; *Ezan et al., 2013*). Because GPSM2-GNAI recruits partners to pull on astral microtubules during mitotic spindle orientation, the authors proposed that GPSM2-GNAI functions similarly and triggers the basal body off-center migration when post-mitotic HC break planar symmetry. However, this hypothesis has not been validated in vivo to date. Studies where *Gpsm2* (*Bhonker et al., 2016*; *Ezan et al., 2013*; *Mauriac et al., 2017*; *Tarchini et al., 2013*), *Gnai3* (*Beer-Hammer et al., 2018*; *Ezan et al., 2013*; *Mauriac et al., 2017*), or *Gnai2*; *Gnai3* (*Beer-Hammer et al., 2018*) were inactivated did not report symmetrical HC. In addition, mouse strains expressing ptxA in vivo also did not produce symmetrical cochlear HC (*Tarchini et al., 2013*; *Tarchini et al., 2016*).

Second, ptx experiments induced striking HC misorientation. In the cochlea, misorientation manifested as a 180° inversion of outer HC in the first and second row (OHC1–2) whereas inner HC (IHC) and OHC3 were much less affected (*Figure 1C*; *Ezan et al., 2013*; *Kindt et al., 2021*; *Tarchini et al., 2013*). In the vestibular system, ptxA expression abrogated the line of polarity reversal and thus the mirror-image HC organization characteristic of macular organs, the utricle and saccule (*Jiang et al., 2017*; *Kindt et al., 2021*). Normal orientation reversal was also lost upon inactivating two endogenous mouse proteins: the transcription factor EMX2 in the maculae (*Jiang et al., 2017*) and the orphan GPCR GPR156 in cochlear OHC1–2 and in the maculae (*Kindt et al., 2021*). Together, these recent studies uncovered an EMX2>GPR156>GNAI signaling cascade that secures a normal pattern of HC orientation by reversing *Emx2*-positive HC. GNAI signals downstream of GPR156 to reverse the orientation of the basal body migration in *Emx2*-positive compared to *Emx2*-negative HC (*Tona and Wu, 2020*) (reviewed in *Tarchini, 2021*). While ptx impact on orientation thus appears to be physiologically relevant, HC misorientation was surprisingly not reported in single *Gnai3* or double *Gnai2*; *Gnai3* mutants (*Beer-Hammer et al., 2018*; *Mauriac et al., 2017*).

In summary, the importance of the GPSM2-GNAI complex for hair bundle development is well established, but multiple discrepancies cast a doubt on whether GNAI proteins also assume earlier polarization roles. Specific questions include whether GNAI proteins participate in the mechanism that pushes the basal body away from the cell center, and in the distinct EMX2>GPR156 mechanism that makes a binary decision on the direction of this push (*Figure 1B*). Furthermore, it remains unclear whether GNAI2 and GNAI3 adopt similar or distinct distributions in HC, and whether GNAI1 or GNAO also participate in these processes.

In this study, we embarked on a systematic analysis of single and combined *Gnai1*, *Gnai2*, *Gnai3*, and *Gnao1* mouse mutants to solve the actual role(s) of GNAI/O proteins during HC differentiation. Our results confirm that GNAI3 is the only GNAI/O protein required for normal hair bundle morphogenesis and normal auditory brainstem response (ABR) thresholds. In absence of GNAI3, GNAI2 can fully compensate at embryonic stages but is not enriched with GPSM2 long enough to ensure normal

hair bundle morphogenesis at postnatal stages. We directly demonstrate that GNAI proteins have two early polarization roles independent of GPSM2 during embryogenesis. In sum, GNAI function is instrumental for HC to (a) break planar symmetry, (b) adopt a proper binary orientation along the PCP axis downstream of EMX2 and GPR156, and (c) elongate and organize stereocilia into a functional hair bundle with GPSM2.

## Results

### A near-comprehensive collection of *Gnai/o* mouse mutants

In order to interrogate the individual and combined roles of all inhibitory G proteins during HC differentiation, we obtained or generated mouse strains to build a collection of single and double *Gnai/o* mutants. Single *Gnai1* and *Gnai3* mutants were derived from the *Gnai1*$^{tm1Lbi}$; *Gnai3*$^{tm1Lbi}$ double mutant strain (hereafter *Gnai1*$^{neo}$; *Gnai3*$^{neo}$) (*Jiang et al., 2002*) by segregating individual mutations upon breeding (see Methods and *Supplementary file 1* for details on all strains). We generated a new constitutive *Gnai2* mutant strain (*Gnai2*$^{del}$) carrying a deletion of exons 2–4 (*Figure 2—figure supplement 1A*; see Methods). Finally, we obtained and derived two *Gnao1* mutant strains: a constitutive inactivation allele where a neomycin cassette disrupts exon 6 (*Gnao1*$^{neo}$) (*Jiang et al., 1998*) and the conditional inactivation allele *Gnao1*$^{tm1c(EUCOMM)Hmgu}$ (hereafter *Gnao1*$^{flox}$). As simultaneous constitutive loss of GNAI1 and GNAI2 was reported as viable (*Plummer et al., 2012*), we established a *Gnai1*$^{neo}$; *Gnai2*$^{del}$ double mutant strain in addition to *Gnai1*$^{neo}$; *Gnai3*$^{neo}$ (*Jiang et al., 2002*). In contrast, double inactivation of *Gnai2; Gnai3* is lethal around embryonic day (E) 10.5, before HC are born (*Gohla et al., 2007*). Consequently, we generated a new *Gnai3*$^{flox}$ strain by flanking exons 2 and 3 with *loxP* sites (*Figure 2—figure supplement 1B*; see Methods). We then generated conditional *Foxg1-Cre; Gnai2*$^{del}$; *Gnai3*$^{flox}$ double mutants where *Gnai3* inactivation occurs as early as E8.5 in the otic vesicle (*Hébert and McConnell, 2000*). Investigating all *Gnai/o* strains in the same genetic background was unrealistic for feasibility and lethality reasons. We reasoned that apical HC development is probably highly constrained and less likely to be influenced by genetic heterogeneity compared to susceptibility to disease, for example.

As two of the three defects directly attributed to GNAI/O dysfunction were only observed using ptx (*Figure 1C*), the *Gnai/o* strains above needed to be compared to a strain expressing ptxA in HC. We used our *Rosa26*$^{LSL-myc:ptxA}$ strain (hereafter *LSL-myc:ptxA*) expressing N-terminal myc-tagged ptxA upon Cre recombination (*Figure 2—figure supplement 1C*; *Tarchini et al., 2016*). Because a related *Rosa26*$^{LSL-ptxAa:myc}$ strain carrying a C-terminal myc tag (*Regard et al., 2007*) caused milder HC misorientation defects than *LSL-myc:ptxA* in the vestibular system (*Jiang et al., 2017*; *Kindt et al., 2021*), we wondered whether the myc tag could weaken toxin activity even perhaps when located N-terminal. Consequently, we generated a new strain, *Rosa26*$^{DIO-ptxA}$ (hereafter *DIO-ptxA*), where untagged ptxA is flanked by double-inverted *lox* sites and flipped from the non-coding to the coding strand upon Cre recombination (*Figure 2—figure supplement 1D*; see Methods) (*Schnütgen et al., 2003*). In *DIO-ptxA, ptxA* expression is driven by a strong artificial CAG promoter, and not by the endogenous *Rosa26* promoter as in previous strains (*Regard et al., 2007*; *Tarchini et al., 2016*). We bred *LSL-myc:ptxA* and *DIO-ptxA* either with *Foxg1*-Cre active in otic progenitors (*Hébert and McConnell, 2000*) or alternatively with *Atoh1-Cre* (*Matei et al., 2005*) to limit GNAI/O inhibition to post-mitotic HC. *Supplementary file 1* summarizes the strains used in this study, their origin, genetic background, and viability.

### Only GNAI3 is required for hair bundle morphogenesis yet GNAI2 participates

To investigate the role of individual GNAI/O proteins in HC development, we imaged hair bundles in 3-week-old mutant and control littermate mice at the mid cochlear position using scanning electron microscopy (SEM). Single *Gnai1* or *Gnai2* mutants as well as double *Gnai1; Gnai2* mutants did not show overt defects in OHC or IHC (*Figure 2A*). In single *Gnai3* mutants by contrast, some OHC hair bundles appeared truncated, and IHC displayed variably shortened row 1 stereocilia as well as supernumerary stereocilia rows (*Figure 2A and B*), as previously described (*Mauriac et al., 2017*). The same defects were observed in double *Gnai1; Gnai3* mutants and, as reported previously, in the *LSL-myc:ptxA* model (*Figure 2A and B*; *Tadenev et al., 2019*). A constitutive (*Gnao1*$^{neo}$) or conditional

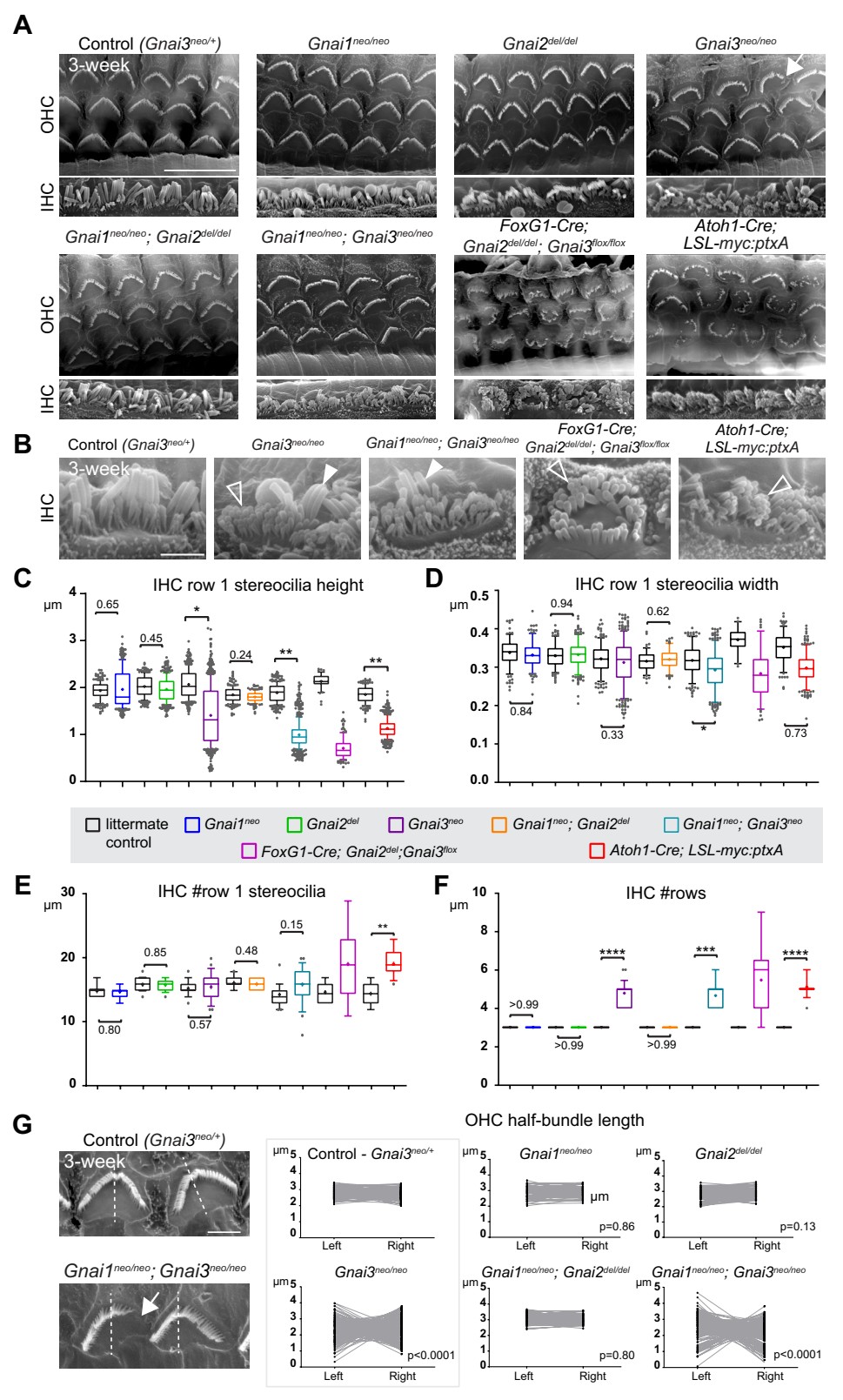

**Figure 2.** Individual GNAI proteins make different contributions to hair bundle development. (**A** and **B**) Scanning electron microscopy (SEM) images of representative OHC (**A**) and IHC (**B**) in 3-week-old animals at the cochlear mid. *Gnai1^neo^*, *Gnai2^del^*, and *Gnai1^neo^; Gnai2^del^* mutants show apparently normal hair bundles in both HC types. In contrast, *Gnai3^neo^* and *Gnai1^neo^; Gnai3^neo^* mutants show defects in both HC types, including truncated hair bundles

*Figure 2 continued on next page*

*Figure 2 continued*

in OHC (arrow), as well as supernumerary rows of stunted (hollow arrowheads) or variable height stereocilia (full arrowheads) in IHC. In addition, in *Foxg1-Cre; Gnai2^del^; Gnai3^flox^* and *Atoh1-Cre; LSL-myc:ptxA* mutants, OHC1–2s are severely misoriented. (**C–F**) Quantification of various hair bundle features in 3-week-old IHC at the cochlear mid. Each mutant strain is compared to littermate controls (in black). At least 3 animals, 17 IHC, and 108 stereocilia are represented per condition, except for *Foxg1-Cre; Gnai2^del^; Gnai3^flox^* where we could only obtain a single adult animal due to postnatal lethality. Nested (hierarchical) t-test sorted by animal; $p<0.0001$****, $p<0.001$***, $p<0.01$**, $p<0.05$*; non-significant p-values are indicated. (**G**) SEM images of representative OHC showing a truncated hair bundle (arrow). Lengths of the left and right wing of the hair bundle were measured and plotted as paired values for the same OHC. A littermate control graph is only shown for *Gnai3* mutants (*Gnai3^neo/+^* controls). Littermate control graphs for the other mutants can be found in *Figure 2—figure supplement 1G*. p-values are for an F-test of variance of pooled left and right wing lengths compared to littermate controls. At least 3 animals and 88 OHC are represented per genotype. Only *Gnai3* and *Gnai1; Gnai3* mutants show truncated hair bundles and a significant p-value ($p<0.05$). Scale bars are 10 μm (**A**) and 2 μm (**B, G**). OHC, outer hair cell; IHC, inner hair cell.

The online version of this article includes the following figure supplement(s) for figure 2:

**Figure supplement 1.** New mouse strains generated, normal apical hair cell morphology in absence of GNAO and control littermate graphs for *Figure 2G*.

---

(*Atoh1-Cre; Gnao1^flox^*) inactivation of *Gnao1* did not produce overt apical HC defects (*Figure 2—figure supplement 1E and F*). Conditional inactivation of *Gnao1* also did not enhance defects in the *Gnai1; Gnai3* mutant background (*Figure 2—figure supplement 1F*).

In addition, *myc:ptxA* expression inverted the orientation of OHC1–2s (*Figure 2A*), as expected since ptxA inactivates the EMX2>GPR156>GNAI signaling cascade that defines HC orientation along the PCP axis (*Kindt et al., 2021*; *Tarchini et al., 2013*; *Tarchini et al., 2016*). However, a defect in HC orientation was not observed in single *Gnai1, Gnai2, Gnai3* mutants, or in double *Gnai1; Gnai2* and *Gnai1; Gnai3* mutants (*Figure 2A*). Despite extensive breeding, we could only obtain one adult animal carrying a double *Gnai2; Gnai3* inactivation due to postnatal lethality (*Foxg1-Cre; Gnai2^del/del^; Gnai3^flox/flox^*; see *Supplementary file 1*). Remarkably, this unique specimen not only recapitulated stereocilia stunting and extra stereocilia rows observed in the *Gnai3^neo^*, *Gnai1^neo^; Gnai3^neo^*, and *LSL-myc:ptxA* models (*Figure 2A and B*), but apparently also OHC1–2 misorientation only observed to date in the *LSL-myc:ptxA* and *Gpr156* mutants (*Figure 2A*; *Kindt et al., 2021*). This result suggests that endogenous GNAI proteins are involved in HC orientation, and that GNAI2 can function in the EMX2>GPR156>GNAI signaling cascade to secure proper OHC1–2 orientation in *Gnai1; Gnai3* double mutants. This point is verified and expanded below when neonate HC orientation is addressed.

To acquire a quantitative view of *Gnai* mutant defects and help comparisons, we first focused on IHC and measured row 1 stereocilia height and width, as well as the number of stereocilia in row 1 and the number of rows in the bundle in all strains (*Figure 2C–F*). This analysis uncovered a *Gnai3^neo^<Gnai1^neo^*; *Gnai3^neo^<LSL-myc:ptxA<Foxg1-Cre; Gnai2^del^; Gnai3^flox^* allelic series along which (a) defects increased in severity, as manifested by increased variability and decreased averages (row 1 height; *Figure 2C*), and (b) new defects appeared (excess row 1 stereocilia only observed in *LSL-myc:ptxA and Gnai2^del^; Gnai3^flox^*; *Figure 2E*). The phenotypic series moved toward increasingly immature-looking hair bundles, and increasingly mimicked severe defects in *Gpsm2* mutants (*Mauriac et al., 2017*; *Tadenev et al., 2019*; *Tarchini et al., 2016*).

To quantify and compare hair bundle truncations in OHC, we measured the length of the half-bundle 'wings' on each side of the central vertex. We limited this analysis to mutant strains where hair bundles retained a recognizable vertex, thus excluding *LSL-myc:ptxA* and *Foxg1-Cre; Gnai2^del^; Gnai3^flox^*. We then plotted paired left and right values for each OHC and tested length variance for each mutant (*Figure 2G*; *Figure 2—figure supplement 1G*). In *Gnai1, Gnai2* and *Gnai1; Gnai2* mutants, the two wings of each hair bundle had similar lengths, and variance was similar to control littermates. In contrast, variable truncation of one wing in *Gnai3* and *Gnai1; Gnai3* mutants resulted in significantly higher length variance compared to controls (*Figure 2G*; *Figure 2—figure supplement 1G*). In both *Gnai3* and *Gnai1; Gnai3* mutants, 49% of hair bundles had wings of more different lengths than the worst OHC outlier in littermate controls.

In conclusion, all GNAI/O proteins are not equally involved in hair bundle morphogenesis, with GNAI3 playing a particularly prominent role. GNAI2 makes a clear contribution since *Gnai3* mutant stereocilia defects dramatically increase in severity when GNAI1 is also absent in *Gnai2; Gnai3* double

mutants. These results confirm previous conclusions (*Beer-Hammer et al., 2018*). In addition, we largely rule out that GNAO is involved in apical HC differentiation. More severe defects in *Gnai1; Gnai3* double mutants compared to *Gnai3* single mutants may indicate that GNAI1 plays a subtle role. However, we cannot rule out differences in genetic background as the underlying cause since *Gnai3^{neo}* is in mixed (29S1/SvImJ; C57BL/6J) and *Gnai1^{neo}; Gnai3^{neo}* is in pure 129S1/SvImJ background (*Supplementary file 1*). GNAI1 thus has at best a minimal role in hair bundle development.

## Only GNAI3 is required for normal auditory brainstem thresholds

To pair morphological defects with auditory function, we next tested ABR in mutants and control littermates at 3–4 weeks of age. Anesthetized animals were presented with pure tone stimuli of decreasing sound pressure intensity (dB sound pressure level [SPL]) at 8, 16, 32, and 40 kHz and ABRs were recorded with a subcutaneous probe (see Methods). Mirroring their overtly normal apical HC morphology, *Gnai1, Gnai2,* and *Gnai1; Gnai2* mutants showed thresholds comparable to littermate controls at all frequencies (*Figure 3A–C*). Similarly, constitutive (*Gnao1^{neo}*) or conditional (*Atoh1-Cre; Gnao1^{flox}*) loss of GNAO did not alter ABR thresholds (*Figure 3—figure supplement 1A and B*). In contrast, *Gnai3* mutants were profoundly deaf at 32 and 40 kHz and displayed significantly elevated thresholds at 8 and 16 kHz compared to littermate controls (*Figure 3D*). *Gnai1; Gnai3* double mutants shared a similar ABR profile as *Gnai3* single mutants, with apparently higher thresholds at 8 and 16 kHz although these two strains were not compared as littermates (*Figure 3D*). Littermate controls for *Gnai1; Gnai3* double mutants were homozygote for *Gnai1^{neo}* (*Gnai1^{neo/neo}; Gnai3^{neo/+}*) and appeared to have higher thresholds than littermate controls for *Gnai3* mutants (*Gnai3^{neo/+}*) at 32 and 40 kHz but not 8 and 16 kHz, whereas loss of GNAI1 did not impact auditory thresholds on its own (*Figure 3A*). These results match possibly more severe hair bundle defects when GNAI1 is inactivated along with GNAI3 (*Figure 2C–F*), and here again, could reflect either a difference in genetic background or a limited role for GNAI1.

In summary, we confirm that GNAI3, but not GNAI2, is essential for proper auditory thresholds, as previously proposed (*Beer-Hammer et al., 2018*). We also clarify that GNAO does not participate. Beer-Hammer and colleagues previously showed that inactivating GNAI2 worsens hearing loss in the *Gnai3* null background, as tested in their *Foxg1-Cre; Gnai2^{flox/flox}; Gnai3^{flox/flox}* adults (*Beer-Hammer et al., 2018*). Using a comparable yet distinct model (*Foxg1-Cre; Gnai2^{del/del}; Gnai3^{flox/flox}*), we were largely unable to obtain young adults and thus could not perform ABR. This suggests that our model is more severely affected, and would probably lack ABRs, similar to *Foxg1-Cre; Gnai2^{flox/flox}; Gnai3^{flox/flox}* (*Beer-Hammer et al., 2018*) and *LSL-myc:ptxA* (*Tarchini et al., 2016*).

## Distribution of individual GNAI proteins in neonate HC

Next, we attempted to define how individual GNAI/O proteins localize in neonate HC. GNAI/O proteins are close paralogs, with mouse and human GNAI1 and GNAI3 most similar in the GNAI group, and GNAO more divergent compared to all GNAI (*Figure 1A*). Validating and using specific antibodies would thus be challenging. Instead, we used one commercial antibody raised against GNAI3 (scbt"GNAI3") and one antibody raised against GNAI2 (pt"GNAI2") and systematically immunolabeled our mouse model collection to tease apart protein-specific behavior.

Both antibodies produced the familiar GNAI pattern of enrichment at the bare zone and at row 1 stereocilia tips (*Figure 4*; arrows and arrowheads, respectively) (*Tarchini et al., 2013*; *Tarchini et al., 2016*). Both antibodies revealed generally normal GNAI enrichment in *Gnai1* (*Figure 4A*), *Gnai2* (*Figure 4B*), and *Gnai1; Gnai2* mutants (*Figure 4C*). This indicates that pt"GNAI2" is not specific for GNAI2 and can also detect GNAI3. In *Gnai3* and *Gnai1; Gnai3* mutants, both antibodies still detected GNAI protein at the bare zone and at stereocilia tips although some HC displayed incomplete enrichment (*Figure 4D and E*; detailed analysis below, *Figure 5*). This indicates that scbt"GNAI3" is not specific for GNAI3 and can also detect GNAI2. Finally, neither antibody detected consistent signal over background in *Gnai2; Gnai3* double mutants where GNAI1 is the only GNAI protein remaining (*Figure 4F*; *Foxg1-Cre; Gnai2^{del/del}; Gnai3^{flox/flox}*). To test experimentally whether these antibodies can detect GNAI1, we first electroporated *Gnai1, Gnai2,* or *Gnai3* constructs in E13.5 inner ears and cultured the cochlea for 6 days before immunolabeling. Overexpressed *Gnai1* was efficiently detected by pt"GNAI2" but not by scbt"GNAI3" whereas, as expected, overexpressed *Gnai3* was detected by scbt"GNAI3" (*Figure 4—figure supplement 1A*). We next used pt"GNAI2" to immunolabel the

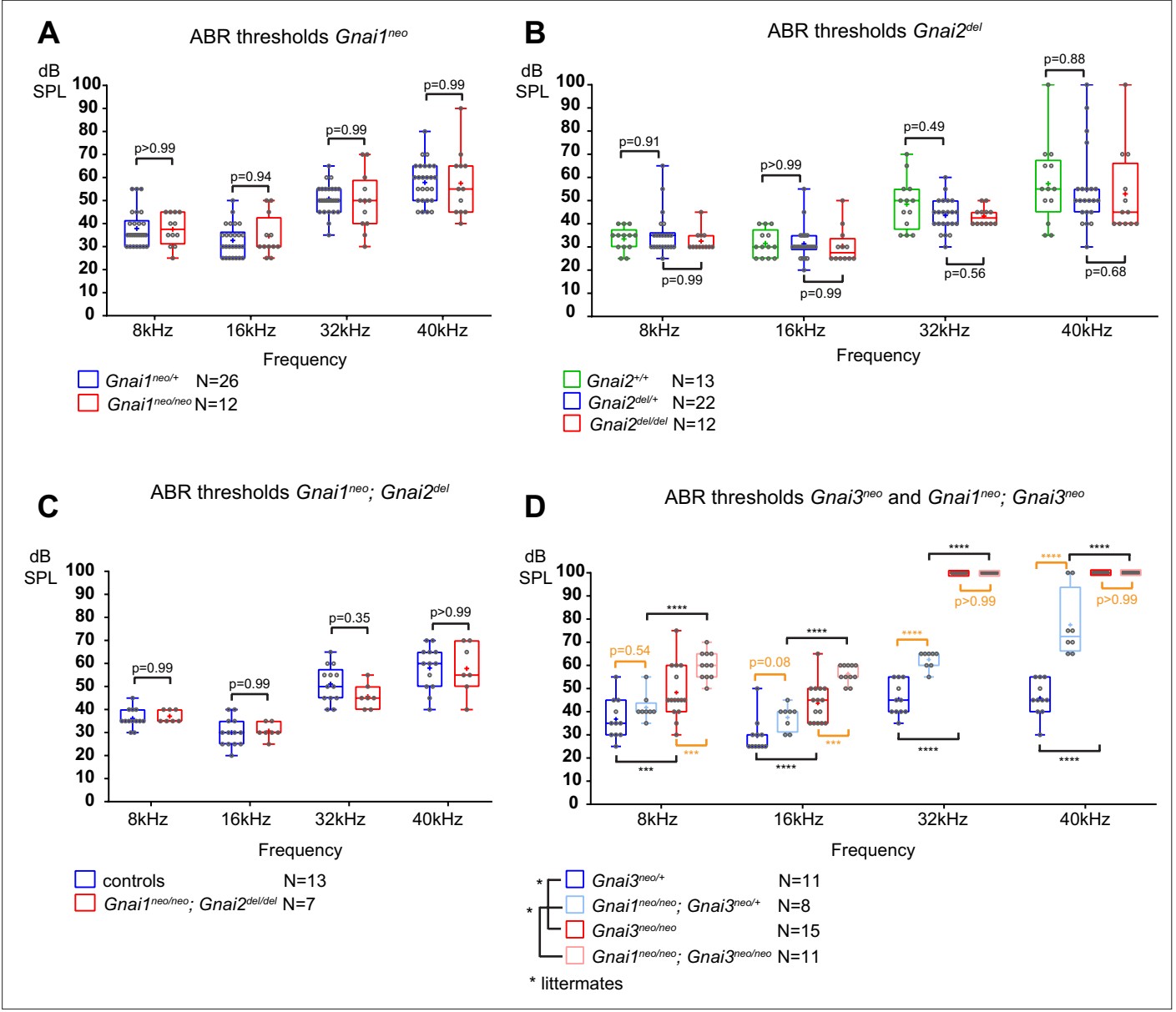

**Figure 3.** Loss of GNAI3 leads to hearing loss most severe at high frequencies. (**A–D**) Auditory brainstem response (ABR) thresholds at 8, 16, 32, and 40 kHz for *Gnai1neo* (**A**), *Gnai2del* (**B**), *Gnai1neo; Gnai2del* (**C**), and *Gnai3neo* and *Gnai1neo; Gnai3neo* (**D**) mutants tested between P21 and P29. Boxplots are framed with 25–75% whisker boxes where exterior lines show the minimum and maximum values, the middle line represents the median, and + represent the mean. A plotted value of 100 dB indicates animals that did not respond to 90 dB stimuli. In (**C**), controls are a pool of *Gnai1+/+; Gnai2del/+*, *Gnai1neo/+; Gnai2+/+*, and *Gnai1neo/+; Gnai2del/+* animals. N indicates the number of animals of both sexes tested. Two-way ANOVA with Sidak's multiple comparison. p<0.0001****, p<0.001***; non-significant p-values are indicated. p-values in orange were obtained comparing non-littermate animals and suggest possibly raised thresholds when GNAI1 is inactivated in addition to GNAI3, or due to a difference in genetic background (see text). kHz, kiloHertz, dB SPL, decibel sound pressure level.

The online version of this article includes the following figure supplement(s) for figure 3:

**Figure supplement 1.** Loss of GNAO does not impact hearing thresholds.

gallblader epithelium where *Gnai1* expression was specifically reported (mousephenotype.org; *LacZ* reporter in *Gnai1tm1a(EUCOMM)Wtsi* strain). Signals were visibly reduced in *Gnai1* mutants compared to littermate controls (*Figure 4—figure supplement 1B*).

Together, these results indicate that (a) GNAI2 and GNAI3 share the same polarized distribution pattern at the bare zone and at stereocilia tips, and (b) GNAI1 is absent or hidden by background signals at the HC apical surface since it can be detected by the pt"GNAI2" antibody in other contexts.

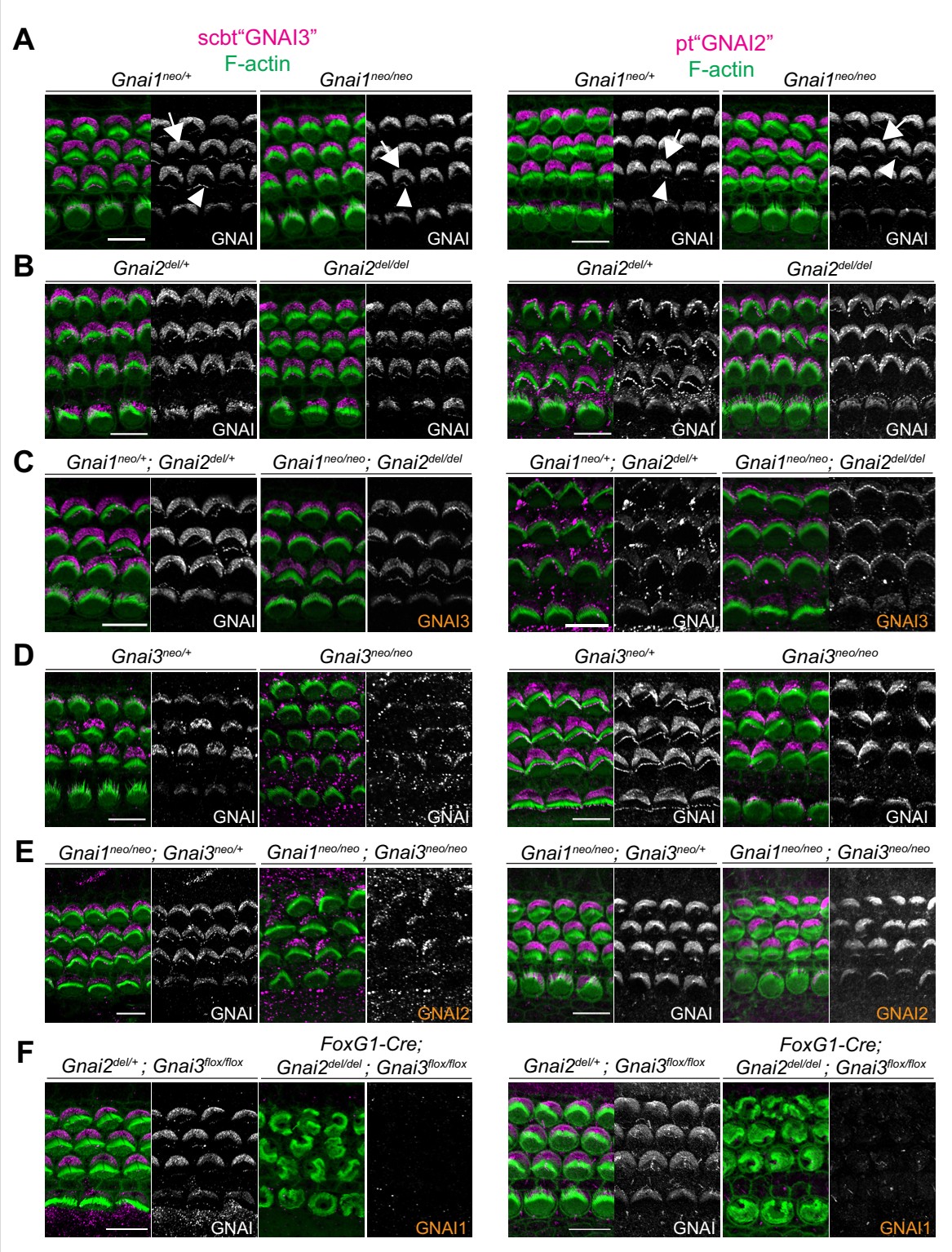

**Figure 4.** Systematic immunolabeling of GNAI proteins in *Gnai* mutant strains. (**A–F**) Two different antibodies (scbt"GNAI3" and pt"GNAI2") were used to label the auditory epithelium at P0-P3. GNAI proteins were detected at the bare zone (arrows) and at stereocilia tips (arrowheads). Neither antibody is specific for its protein target, as scbt"GNAI3" is able to detect GNAI2 (**E**) and pt"GNAI2" is able to detect GNAI3 (**C**). Note how no apical GNAI signal is visible with either antibody in *Gnai2; Gnai3* double mutants (**F**), showing that GNAI1 is not enriched apically in HC (see *Figure 4—figure supplement 1A and B* for evidence that pt"GNAI2" detects GNAI1). When the identity of the GNAI protein detected is unambiguous based on genotype, it is made explicit in orange. Scale bars are 10 μm.

*Figure 4 continued on next page*

*Figure 4 continued*

The online version of this article includes the following figure supplement(s) for figure 4:

**Figure supplement 1.** GNAI1 detection by Proteintech GNAI2 antibody, GNAI protein distribution in *Gnao1* mutants, and lack of evidence for GNAO enrichment at the hair cell apex.

Loss of GNAO in the *Gnao1^neo* or *Atoh1-Cre; Gnao1^flox* models did not alter signals obtained with scbt"GNAI3" (*Figure 4—figure supplement 1C and D*). Moreover, a GNAO antibody produced unpolarized apical signals that proved unspecific since they were unchanged in *Atoh1-Cre; Gnao-1^flox/flox* mutants (*Figure 4—figure supplement 1E*). Thus, GNAO likely does not contribute to HC polarization, as also suggested by normal hair bundles and normal ABR thresholds in *Gnao1* mutants.

## GNAI2 only partially spans the bare zone and stereocilia tips and partially rescues hair bundle development in postnatal HC lacking GNAI3

We next took a closer look at incomplete GNAI enrichment in *Gnai3* and *Gnai1; Gnai3* mutant HC (*Figure 4D and E*). In both models, we observed an identical outcome at the P0 cochlear base where remaining GNAI was unable to fully and consistently occupy the HC sub-domains when GNAI3 was missing (*Figure 5A and B*; bare zone, arrows; stereocilia tips, arrowheads). In *Gnai1; Gnai3* double mutants, the GNAI protein detected is by default GNAI2, and this is likely also true in single *Gnai3* mutants since GNAI1 is not observed in HC (*Figure 4F*). Because in all cases GPSM2 co-localized with GNAI2 (*Figure 5A and B*), these results demonstrate that both GNAI2 and GNAI3 can form a complex with GPSM2 in HC. Intriguingly, the absence of GPSM2-GNAI2 on one side of the bare zone at P0 appeared to coincide with its absence at stereocilia tips on the corresponding side (*Figure 5A and B*). We thus quantified GNAI2 intensity at the bare zone and at stereocilia tips in half-OHC in *Gnai1; Gnai3* mutants. While as expected control OHC showed little variation in GNAI enrichment in either compartment (*Figure 5—figure supplement 1A*), mutant OHC showed highly variable signals that were significantly correlated between the bare zone and tips in the same half-OHC (*Figure 5C*). Analyzing HC at different stages and tonotopic positions clarified that GNAI2 is progressively unable to compensate for missing GNAI3. At E18.5, the GPSM2-GNAI2 complex could still occupy the totality of the bare zone in *Gnai1; Gnai3* mutants (*Figure 5—figure supplement 1B*), suggesting that GNAI2 fully compensates for the loss of GNAI3 in embryonic HC. At later stages of HC differentiation, partial compensation by GNAI2 observed at P0 (*Figure 5A–C*) evolved into a lack of compensation by P6 at the cochlear base, where GNAI2 was no longer detected consistently at the tips of stunted stereocilia in mutant IHC (*Figure 5D*). In contrast, less mature P6 IHC at the cochlear mid position retained partial GNAI2 enrichment correlated between the bare zone and tips (*Figure 5E*), as seen at P0 (*Figure 5A and B*).

The progressive inability of GNAI2 to cover for GNAI3 in individual HC helps explain the unique profile of apical defects in *Gnai3* and *Gnai1; Gnai3* mutants. Loss of GPSM2-GNAI2 in one wing of the OHC hair bundle led to stereocilia degeneration by P8 (*Figure 5F*). One-sided loss of global GNAI function at stereocilia tips is thus likely the origin of truncated hair bundle wings observed in adults *Gnai3* and *Gnai1; Gnai3* mutant OHC (*Figure 2G*). We divided the P8 OHC apical surface in two halves based on the position of the basal body at the hair bundle vertex, and measured the length of each hair bundle wing as well as the apical surface area in the same HC half in *Gnai1; Gnai3* mutants (*Figure 5F*). We found a significant correlation between these two values (*Figure 5G*; control graph in *Figure 5—figure supplement 1C*), providing further evidence that loss of stereocilia prompts a corresponding loss of flat HC surface area on the same OHC side (*Etournay et al., 2010*). Finally, we asked whether in time GNAI2 is lost in all stereocilia and along the entire cochlea in *Gnai1; Gnai3* mutants. This proved not to be the case, as GNAI2 could still be detected at stereocilia tips in P28 OHC and IHC, although in low and variable amounts compared to GNAI tip signals in littermate controls (*Figure 5—figure supplement 1D and E*).

In conclusion, GNAI2 could provide a low dose of GNAI protein at stereocilia tips when GNAI3 is missing and preserve elongation and height to some extent. Variable GNAI2 amounts thus likely explain why IHC stereocilia have variably reduced heights in absence of GNAI3 (*Figure 2B and C*; *Figure 5D and E*), unlike in *Gpsm2* or *LSL-myc:ptxA* mutants where they are more uniformly stunted

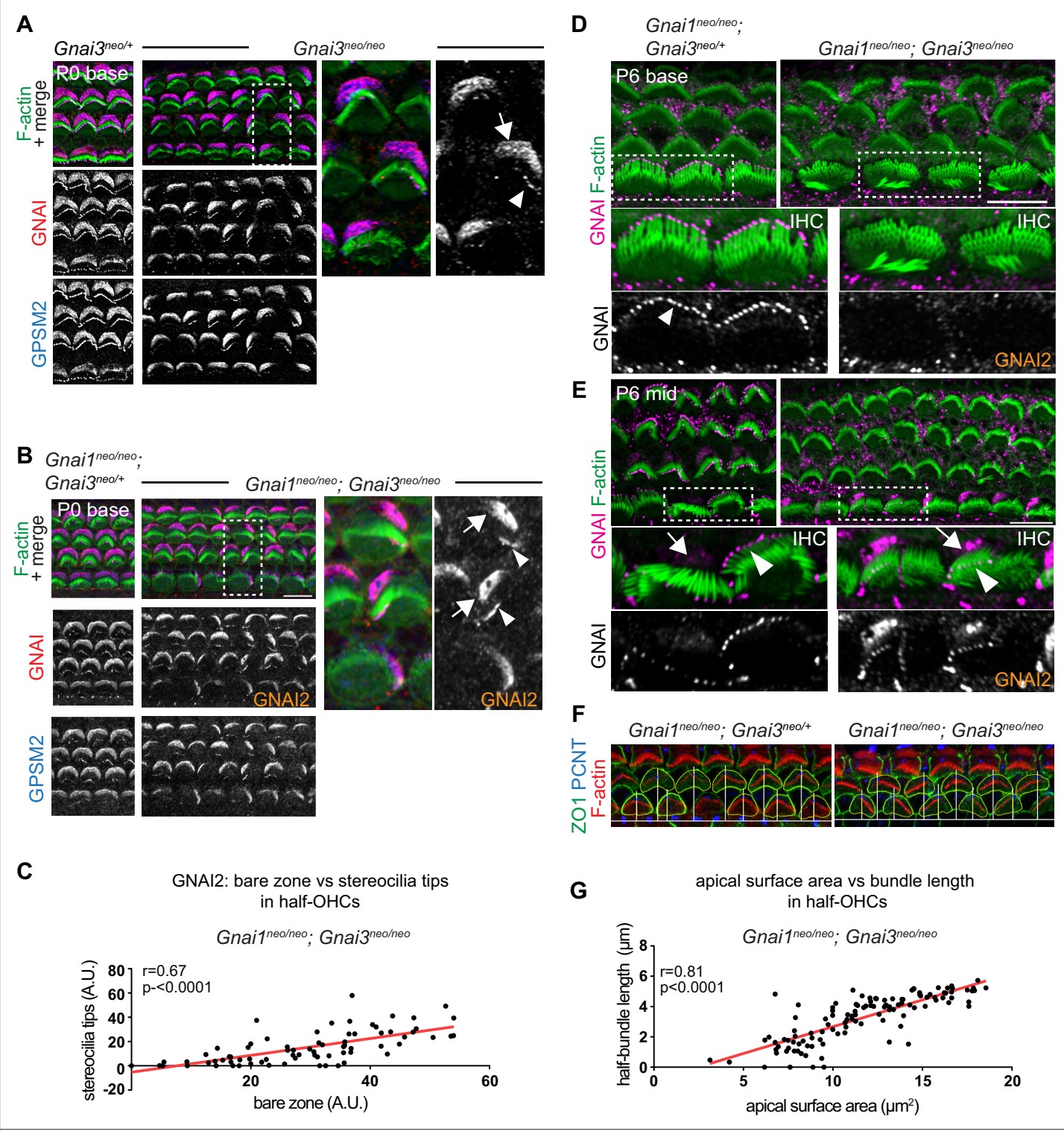

**Figure 5.** GNAI2 only partially rescues loss of GNAI3 in individual postnatal hair cells. (**A** and **B**) GNAI (pt"GNAI2" antibody; see *Figure 4*) and GPSM2 co-immunolabeling in P0 *Gnai3^neo* (**A**) and *Gnai1^neo*; *Gnai3^neo* (**B**) animals at the cochlear base. Boxed regions are magnified on the right. In both mutants, incomplete GNAI patterns are observed at the bare zone (arrow) and stereocilia tips (arrowheads). Remaining GNAI signals must reflect GNAI2 in (**B**). (**C**) Correlation plot of GNAI signal intensity at the bare zone and tips in half-OHC at the P2 cochlear mid. Presence or absence of GNAI is remarkably correlated spatially between bare zone and tips in the same half-OHC. N=3 animals, n=37 OHC, Pearson correlation with best fit (red line; plot for control littermates can be found in *Figure 5—figure supplement 1A*). (**D** and **E**) GNAI (pt"GNAI2" antibody) immunolabeling in P6 *Gnai1^neo*; *Gnai3^neo* animals. Boxed IHC regions are magnified below. Loss of GNAI2 progresses with HC differentiation, with largely absent IHC signals

*Figure 5 continued on next page*

*Figure 5 continued*

at the P6 cochlear base (**D**) but partial rescue on one side of the cell at the P6 mid (**E**), as observed at the cochlear base at P0 (**A** and **B**). (**F**) ZO1 (apical junctions) and pericentrin (PCNT; basal body) immunolabeling in P8 OHC (maximum projection). The position of the basal body is used to determine the vertex (middle) of the original hair bundle and to draw a radial line separating each OHC into two halves. (**G**) The length of each hair bundle wing (y axis) is graphed in relation to the corresponding apical surface area (x axis) in the same half-OHC. Truncated OHC wings correlate with reduced apical membrane area on the same side. N=3 animals, n=58 OHC, Pearson correlation with best fit (red line; plot for control littermates can be found in *Figure 5—figure supplement 1C*). AU, arbitrary unit; IHC, inner hair cell; OHC, outer hair cell. Scale bars are 10 μm.

The online version of this article includes the following figure supplement(s) for figure 5:

**Figure supplement 1.** GNAI2 fully rescues loss of GNAI3 at embryonic stages and is still detected in low amounts at stereocilia tips in adult hair cells lacking GNAI3.

---

(*Beer-Hammer et al., 2018*; *Mauriac et al., 2017*; *Tadenev et al., 2019*; *Tarchini et al., 2016*). Loss of GNAI2 signals that coincides at the bare zone and at stereocilia tips on the same HC side adds to previous evidence suggesting that bare zone enrichment is essential for GPSM2-GNAI trafficking to adjacent row 1 stereocilia (*Akturk et al., 2022*; *Jarysta and Tarchini, 2021*; *Tarchini et al., 2016*).

## Combined loss of GNAI2 and GNAI3 delays and de-polarizes bare zone expansion with drastic consequences on stereocilia distribution

Since GNAI2 and GNAI3 show functional redundancy and are the most important GNAI/O proteins for hair bundle differentiation, we next focused on characterizing early HC development in *Gnai2; Gnai3* double mutants (*Foxg1-Cre; Gnai2^del/del^; Gnai3^flox/flox^*) and comparing defects to those observed previously with ptx. Unlike at adult stages, *Gnai2; Gnai3* double mutants were obtained in close to Mendelian proportions at P0 (see *Supplementary file 1*). For this goal, we used the new *DIO-ptxA* allele in case the myc tag hindered ptxA activity in the *LSL-myc:ptxA* allele. We first validated the new *Gnai3^flox^* and *DIO-ptxA* alleles (*Figure 2—figure supplement 1B and D*). As expected, GNAI signals at the bare zone and stereocilia tips were normal in *Gnai3^flox/flox^* homozygotes but showed a distinctive incomplete pattern along with stunted stereocilia upon Cre recombination (*Figure 6—figure supplement 1A*), as in constitutive mutants (*Figure 5A*). In fact, the new *DIO-ptxA* model produced identical apical HC defects to the earlier *LSL-myc:ptxA* strain in single HC (*Kindt et al., 2021*; *Tarchini et al., 2016*; *Figure 6—figure supplement 1B*). The fraction of inverted OHC1 in the *DIO-ptxA* model was lower than in the *LSL-myc:ptxA* model when using the post-mitotic HC driver *Atoh1-Cre* (*Figure 6—figure supplement 1C*) but identically encompassed 100% of OHC1 with the early *Foxg1-Cre* driver (*Figure 6—figure supplement 1D*). Similar apical HC defects between strains indicate that the N-terminal myc tag and mild *Rosa26* promoter do not limit ptxA activity in *LSL-myc:ptxA*. For comparison purposes, we used the *Foxg1-Cre* driver to inactivate GNAI2/GNAI3 and to express ptxA, the same driver used by *Beer-Hammer et al., 2018*.

Inactivating GNAI2 and GNAI3 abolished the bare zone at E17.5 based on abnormally uniform F-actin signals at the HC surface compared to controls where low F-actin matched GNAI signals (*Figure 6A and B*; arrows point to the bare zone). In contrast, E17.5 *DIO-ptxA* HC had developed a distinct bare zone (*Figure 6C*, arrows). Measuring its surface area by HC type confirmed that the bare zone was virtually absent in *Gnai2; Gnai3* double mutants but only trended as reduced in ptxA-expressing OHC (*Figure 6D*). By P0, most *Gnai2; Gnai3* mutant HC at the cochlear base had developed a region lacking microvilli or stereocilia, but its position at the apical surface was highly irregular, complementing extremely dysmorphic hair bundles (*Figure 6E and F*; arrows point to bare regions). In contrast, P0 *DIO-ptxA* HC displayed largely coherent hair bundles and a polarized bare zone (*Figure 6G*, arrows). Quantifications revealed a significantly reduced bare area in P0 *Gnai2; Gnai3* mutants compared to littermate controls (*Figure 6H*), showing that bare zone emergence and expansion is greatly delayed and deregulated in this model (compare P0 in *Figure 6H* and E17.5 in *Figure 6D*). In P0 *DIO-ptxA* HC, the bare zone surface area was largely comparable to controls, suggesting that a slight delay in expansion is progressively corrected in time in this model (compare P0 in *Figure 6H* and E17.5 in *Figure 6D*). These results best illustrate to date the importance of GPSM2-GNAI for bare zone emergence, expansion, and polarized positioning. While both mutant models consistently delay bare zone expansion, differences in timing and severity suggest that ptxA is not as effective as the *Gnai2; Gnai3* double mutant to inhibit GPSM2-GNAI function. This is further underscored by severe stereocilia distribution defects observed in *Gnai2; Gnai3* but not *DIO-ptxA*

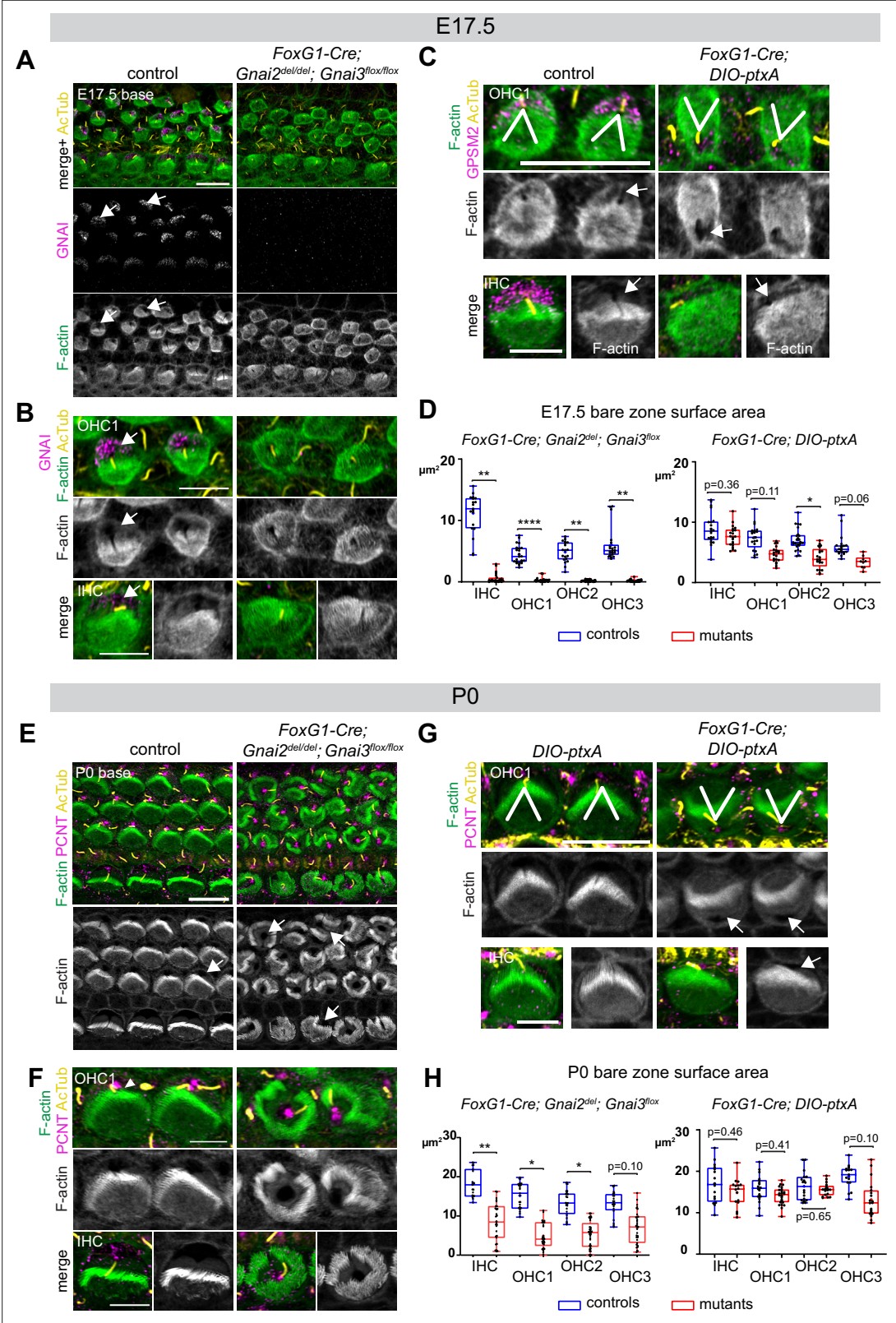

**Figure 6.** Delayed bare zone expansion and severely dysmorphic hair bundles in absence of GNAI2 and GNAI3. (**A** and **B**) GNAI (scbt"GNAI3" antibody, see *Figure 4*) and acetylated tubulin (AcTub; kinocilium) co-immunolabeling at the embryonic day (E) 17.5 cochlear base. Note how F-actin labeling (phalloidin) reveals a polarized bare zone marked by GNAI (arrows) in control but not in *Gnai2; Gnai3* double mutants. (**C**) GPSM2 and AcTub co-immunolabeling at the E17.5 cochlear base. In contrast to *Gnai2; Gnai3* double mutants (**A** and **B**), *Foxg1-Cre; DIO-ptxA* mutants have a polarized

*Figure 6 continued*

bare zone (arrows) despite OHC1–2 adopting a reversed orientation (V brackets indicate OHC1 orientation). GPSM2 marks the bare zone in controls and is reduced in mutants. (**D**) Graphs of bare zone surface area in E17.5 hair cell (HC) at the cochlear base. *Foxg1-Cre; Gnai2^del^; Gnai3^flox^*: controls (*Gnai2^del/+^; Gnai3^flox/+^* and *Gnai2^del/+^; Gnai3^flox/flox^*) N=3 animals, n=19 IHC, 23 OHC1, 19 OHC2, 21 OHC3; mutants N=3, n=23 IHC, 24 OHC1, 23 OHC2, 24 OHC3. *Foxg1-Cre; DIO-ptxA*: controls (Cre-negative *DIO-ptxA*) N=3, n=18 IHC, 18 OHC1, 21 OHC2, 18 OHC3; mutants N=3, n=21 IHC, 20 OHC1, 21 OHC2, 9 OHC3. (**E–G**) Pericentrin (PCNT) and AcTub co-immunolabeling at P0 at the cochlear base. Unlike at E17.5 (**A, B, D**), most P0 *Gnai2; Gnai3* double mutant HC have a bare region (**E** and **F**, arrows). This bare region is unpolarized and its abnormal shape reflects aberrant stereocilia distribution. In sharp contrast, ptxA mutants have normally shaped hair bundles and bare zones despite OHC1–2 adopting a reversed orientation (**G**). (**H**) Graphs of bare zone surface area in P0 HC at the cochlear base. *Foxg1-Cre; Gnai2^del^; Gnai3^flox^*: controls (*Gnai2^del/+^; Gnai3^flox/+^, Gnai2^del/+^; Gnai3^flox/flox^* and *Foxg1-Cre; Gnai2^del/+^; Gnai3^flox/+^*) N=3, n=19 IHC, 23 OHC1, 21 OHC2, 21 OHC3; mutant N=3, n=21 IHC, 22 OHC1, 24 OHC2, 24 OHC3. *Foxg1-Cre; DIO-ptxA*: controls (Cre-negative *DIO-ptxA*) N=3, n=15 IHC, 23 OHC1, 18 OHC2, 15 OHC3; mutants N=3, n=16 IHC, 24 OHC1, 18 OHC2, 19 OHC3. (**D, H**) Nested (hierarchical) t-test sorted by animal; p<0.0001****, p<0.01**, p<0.05*; non-significant p-values are indicated. All ptxA samples are heterozygotes (*Rosa26^DIO-ptxA/+^*). Scale bars are 10 µm (**A, C** [OHC], **E, G** [OHC]) and 5 µm (**B, C** [IHC], **F, G** [IHC]). IHC, inner hair cell; OHC, outer hair cell.

The online version of this article includes the following figure supplement(s) for figure 6:

**Figure supplement 1.** Validation of the *Gnai3^flox^* and *DIO-ptxA* mouse strains.

mutants. Severe stereocilia distribution defects were not observed in *Beer-Hammer et al., 2018* either, suggesting that our *Foxg1-Cre; Gnai2^del^;Gnai3^flox^* mouse model achieves a further loss of GNAI function compared to their *Foxg1-Cre; Gnai2^flox^;Gnai3^flox^* model.

As reported previously (*Kindt et al., 2021*; *Tarchini et al., 2013*; *Tarchini et al., 2016*), the orientation of OHC1–2 expressing ptxA was inverted compared to controls (*Figure 6C and G*). Our single *Gnai2; Gnai3* adult mutant sample also appeared to have inverted OHC1 based on hair bundle morphology (*Figure 2A*). However, HC orientation proved challenging to assess in neonate *Gnai2; Gnai3* mutants due to highly dysmorphic hair bundles (*Figure 6A, B, E, and F*). We thus used acetylated tubulin (AcTub) and pericentrin (PCNT) as markers for the kinocilium and the basal body, respectively. This showed that in *Gnai2; Gnai3,* but not in *DIO-ptxA* mutants, the basal body and kinocilium were often in an approximately central position surrounded partially or entirely by stereocilia (*Figure 6E and F*). Rounded perinatal hair bundles are a hallmark of HC where the early off-center migration of the basal body, hence symmetry breaking, is defective.

To distinguish symmetry breaking from HC orientation, we next used the position of the basal body at the base of the kinocilium to derive both HC eccentricity and HC orientation. This strategy helped compare the *Gnai2; Gnai3* and *DIO-ptxA* mouse models, and identified two early functions for GNAI before its association with GPSM2 for stereocilia elongation.

## GNAI proteins drive the off-center migration of the basal body

We used the position of the basal body to measure HC eccentricity as a metric for cytoskeleton asymmetry. Eccentricity was calculated as a ratio reporting how far away from the cell center the basal body was positioned (see schematic in *Figure 7*). A perfectly symmetrical HC with a central basal body would thus have an eccentricity close to 0 whereas a ratio close to 1 would indicate that the off-center basal body is juxtaposed to the apical junction. Because HC maturation progresses in time and along the tonotopic axis (cochlear apex to base gradient of increasing maturity), we measured eccentricity at E17.5 (base and mid cochlear position) and at P0 (base, mid, and apex) to understand how eccentricity progressed at the HC population level.

Control HC at E17.5 had already broken symmetry at the mid position, with eccentricity averaging ~0.5 and increasing at the more mature base position (*Figure 7A*). At E17.5 mid and base positions, *Gnai2; Gnai3* mutant HC showed a significantly reduced eccentricity, suggestive of a delay in the off-center migration of the basal body. We arbitrarily defined 0.25 as a threshold below which HC were considered near-symmetrical. Symmetrical HC were not observed in controls but represented up to ~29% of IHC and ~34% of OHC in *Gnai2; Gnai3* mutants at E17.5 (detailed in *Figure 7A* by stage, position, and HC type; red highlights). By P0, the proportion of symmetrical cells in mutants had decreased at all positions for OHC (3.3–15.2%), but remained high or increased for IHC (21.5–42.4%; *Figure 7A*).

The situation was radically different in the *DIO-ptxA* model. First, no symmetrical HC was observed at any stage or position in mutants (*Figure 7B*). In the *DIO-ptxA* model, the distribution of eccentricity values in IHC was generally similar to controls. By contrast, OHC eccentricity trended as higher in

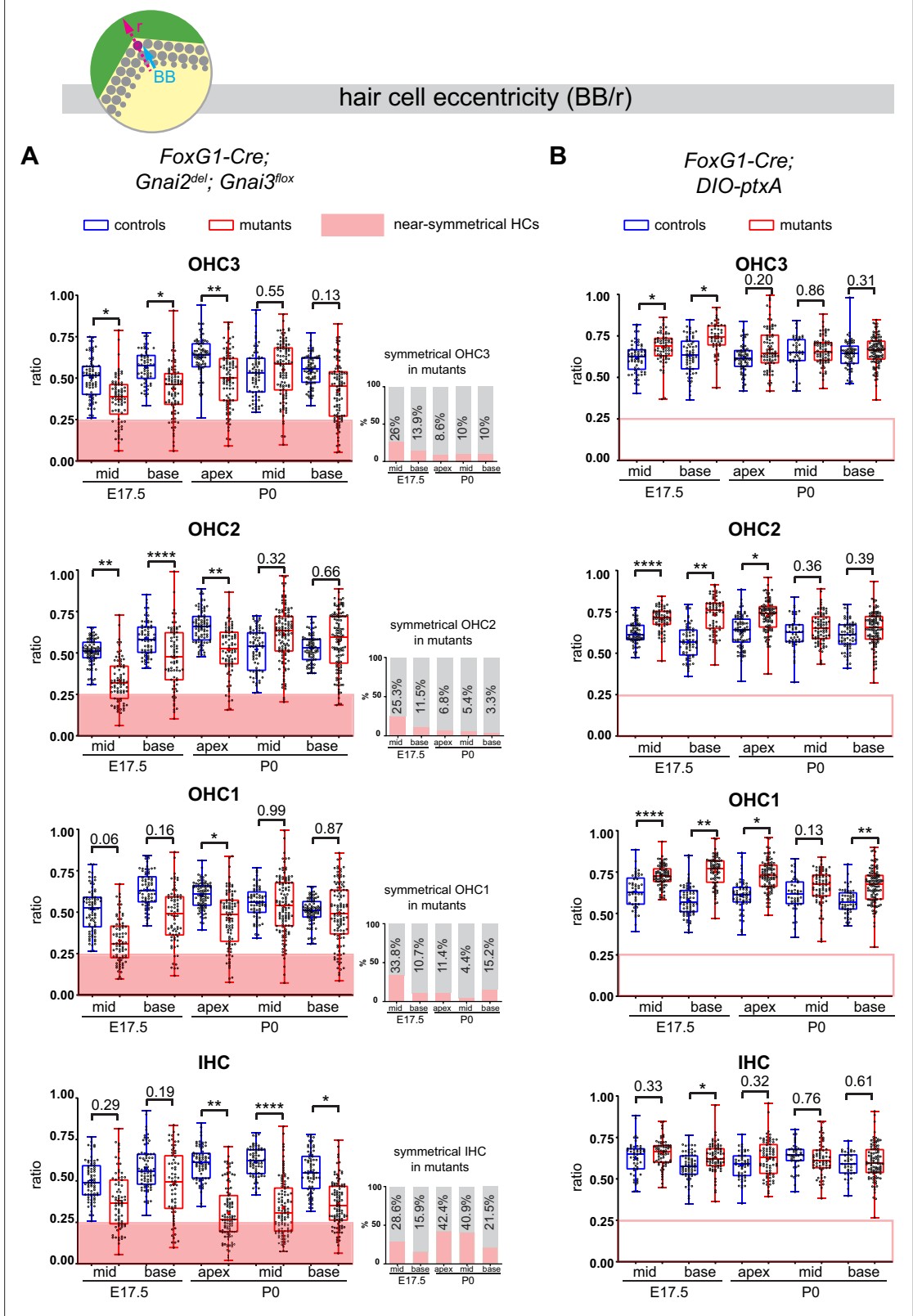

**Figure 7.** Loss of GNAI2 and GNAI3 provokes hair cell (HC) eccentricity defects absent in *ptxA* mutants. (**A** and **B**) Graphs of HC eccentricity representing the position of the basal body as a ratio of the radius (BB/r, top diagram). Data cover embryonic day (E) 17.5 mid and base and P0 apex, mid and base cochlear positions for each HC type. HC were considered near-symmetrical when their eccentricity ratio was lower than 0.25 (red zone). Only *Foxg1-Cre; Gnai2^del^; Gnai3^flox^* mutants harbor symmetrical HC. Their proportion is indicated in the bar graphs on the right (**A**). Overall, the

*Figure 7 continued on next page*

*Figure 7 continued*

proportion of symmetrical cells tends to decrease in maturing outer HC (OHC) but remains high or increases in inner HC (IHC). At least 3 animals and 39 cells per HC type are represented for each stage, cochlear position, and genotype. Controls for *Foxg1-Cre; Gnai2^{del/del}; Gnai3^{flox/flox}* are *Gnai2^{del/+};* *Gnai3^{flox/+}, Gnai2^{del/+}; Gnai3^{flox/flox}, Foxg1-Cre; Gnai2^{del/+}; Gnai3^{flox/+}*, and *Foxg1-Cre; Gnai2^{del/+}; Gnai3^{flox/flox}*. Controls for *Foxg1-Cre; DIO-ptxA* are Cre-negative *DIO-ptxA* heterozygotes. Nested (hierarchical) t-test sorted by animal; p<0.0001****, p<0.01**, p<0.05*; non-significant p-values are indicated.

*DIO-ptxA* compared to littermate controls, with the difference becoming less significant with OHC differentiation (*Figure 7B*). Defective bare zone expansion (*Figure 6D and H*) can help explain the distinct eccentricity defects in the *DIO-ptxA* (transiently higher eccentricity) and *Gnai2; Gnai3* (lower eccentricity) mouse models. The position of the basal body is the sum of two opposite movements (*Figure 1B*): (a) the early off-center migration that brings the basal body in the vicinity of the lateral junction, and (b) the subsequent relocalization toward the cell center upon bare zone expansion that brings the basal body in contact with the forming hair bundle (*Tarchini et al., 2013*). In *DIO-ptxA*, transiently increased eccentricity in OHC is likely the outcome of a normal off-center basal body migration combined with delayed bare zone expansion (*Figure 6D and H*). In *Gnai2; Gnai3* mutants by contrast, decreased eccentricity in OHC (*Figure 7A*) stems from defective off-center migration of the basal body combined with an initially absent (E17.5), and then reduced (P0), bare region (*Figure 6D and H*). When an unpolarized bare region eventually emerges in *Gnai2; Gnai3* mutants (*Figure 6H*), it impacts basal body position without directionality, leading to highly variable eccentricity values compared to controls (*Figure 7A*), and thus variably dysmorphic hair bundles. The apparent increase in symmetrical IHC over time in *Gnai2; Gnai3* mutants (*Figure 7A*) may result from the delayed expansion of the bare region, which will relocalize the basal body centrally in a proportion of IHC.

In conclusion, symmetry breaking (*Figure 7*), bare zone emergence and expansion, and stereocilia distribution (*Figure 6*) are all severely impaired in *Gnai2; Gnai3* mutants. In ptxA mutants in contrast, symmetry breaking occurs normally, bare zone expansion is only delayed, and stereocilia distribution is less affected. It follows that ptxA does not achieve a loss of GNAI2/GNAI3 function as extensive as in *Foxg1-Cre; Gnai2; Gnai3* mutants. This is probably because ptxA only downregulates and does not inactivate GNAI/O proteins, and because ptxA substrates also include GNAI1 and GNAO that could be active in other contexts than polarization in HC.

## GNAI proteins orient HC laterally

GNAI proteins were proposed to signal downstream of the GPR156 receptor to regulate HC orientation in auditory and vestibular organs (*Jiang et al., 2017*; *Kindt et al., 2021*). In HC expressing the transcription factor EMX2, GPR156-GNAI reverses the orientation of the off-center basal body migration, and thus HC orientation, compared to *Emx2*-negative HC (*Tona and Wu, 2020*). To date, however, the HC misorientation profile observed in *Emx2* or *Gpr156* mutants was only recapitulated using ptxA (*Figure 1C*; *Kindt et al., 2021*). If ptxA-based orientation defects are physiologically relevant to GNAI function, they might be observed in *Gnai2; Gnai3* mutants as well. As mentioned above, a caveat to test this assumption is that highly dysmorphic hair bundles in *Gnai2; Gnai3* mutants obscure HC orientation.

To circumvent this limitation, we first excluded near-symmetrical HC in the *Gnai2; Gnai3* mutant datasets since an asymmetric cytoskeleton is pre-requisite for HC to exhibit a defined orientation (excluded: eccentricity <0.4 at E17.5 and <0.25 at P0). Next, we used the position of the off-center basal body to infer the orientation of the remaining asymmetric HC, ignoring the shape of the hair bundle. We measured the angle formed by a vector running from the HC center to the basal body relative to the cochlear longitudinal axis (α, see schematic in *Figure 8*). Angles were plotted in circular histograms where 0° points to the cochlear base and 90° to the lateral edge. At E17.5 at the cochlear mid position, *DIO-ptxA* showed the graded pattern of OHC misorientation observed as early as these cells break planar symmetry (*Kindt et al., 2021*), with inverted OHC1 and OHC2 and imprecisely oriented IHC and OHC3 (*Figure 8A*). *DIO-ptxA* OHC1–2 maintained this misorientation profile at the E17.5 base and at P0 (*Figure 8B–D*) as well as in adults (*Figure 2A*).

At the E17.5 cochlear mid, many but not all OHC1 were strikingly inverted as well in *Gnai2; Gnai3* mutants (*Figure 8A*). The majority of more mature OHC1 at the E17.5 cochlear base were inverted (*Figure 8B*), as were P0 OHC1 at all positions (*Figure 8C and D*; *Figure 8—figure supplement 1A*). Loss of GNAI2 and GNAI3 can thus recapitulate inverted OHC1–2 observed in the *DIO-ptxA* model

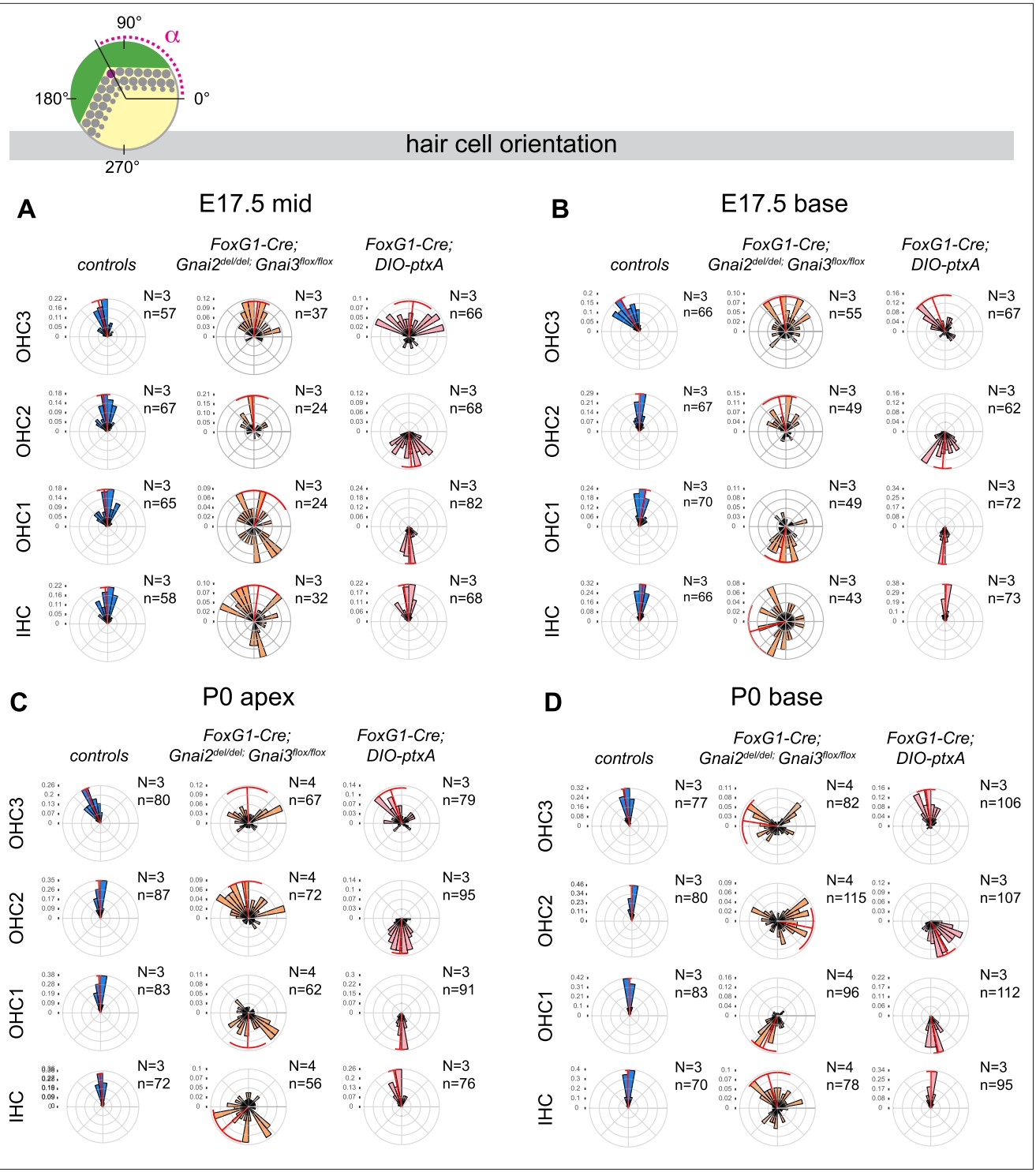

**Figure 8.** Loss of GNAI2 and GNAI3 recapitulates hair cell (HC) orientation defects observed in *ptxA* mutants. (**A–D**) Circular histograms showing HC orientation (α) based on the position of the basal body (purple dot in top diagram) at the stage and cochlear position indicated. 0° is toward the cochlear base and 90° is lateral. HC represented have an eccentricity greater than 0.4 in **A** and **B** (embryonic day [E] 17.5), and 0.25 in **C** and **D** (P0) (see *Figure 7*). Histograms show frequency distribution (10° bins) and red radial lines and arcs respectively indicate circular mean and circular mean deviation. In control cochleae, HC are tightly oriented laterally (90°) except for OHC3 that show a slight bias toward the cochlear apex (180°). As reported previously, ptxA expression inverts OHC1 and OHC2, and results in imprecise lateral orientation of OHC3. This phenotype is recapitulated in *Gnai2; Gnai3* double mutants with a delay (compare least mature E17.5 mid (**A**) and most mature P0 base (**D**)). Inner HC (IHC) also show severe misorientation in *Gnai2; Gnai3* double mutants, unlike in *ptxA* mutants. n, HC number in N=3–4 animals. Controls for *Foxg1-Cre; Gnai2^{del/del}; Gnai3^{flox/*

*Figure 8 continued on next page*

*Figure 8 continued*

*flox* are *Gnai2del/+; Gnai3flox/+*, *Gnai2del/+; Gnai3flox/flox*, *Foxg1-Cre; Gnai2del/+; Gnai3flox/+* and *Foxg1-Cre; Gnai2del/+; Gnai3flox/flox*. Data at the P0 cochlear mid position can be found in *Figure 8—figure supplement 1A*. Histograms for littermate controls of *Foxg1-Cre; DIO-ptxA* mutants can be found in *Figure 8—figure supplement 2*.

The online version of this article includes the following figure supplement(s) for figure 8:

**Figure supplement 1.** Loss of GNAI2 and GNAI3 recapitulates hair cell (HC) orientation defects observed in *ptxA* mutants.

**Figure supplement 2.** Littermate control histograms for *ptxA* mutants.

with what appears to be a delay. In contrast, however, OHC2 were generally oriented laterally (90°) at E17.5 and at the P0 apex and mid in *Gnai2; Gnai3* mutants (*Figure 8A–C*; *Figure 8—figure supplement 1A*). OHC2 only adopted inverted orientation characteristic of the *DIO-ptxA* model at the P0 base where HC are more mature (*Figure 8D*). As OHC2 mature later than OHC1 (*Anniko, 1983*), this again suggests a delay where GPR156-GNAI signaling can initially reverse OHC1–2 normally, but where a lateral orientation cannot be maintained and OHC1–2 ultimately become inverted as in the *DIO-ptxA* model.

*Foxg1-Cre* is expressed at the otocyst stage in cochlear progenitors (*Hébert and McConnell, 2000*) and globally downregulates GNAI/O proteins early on in the *DIO-ptxA* model. However, *Foxg1-Cre*-mediated deletion at the *Gnai3* locus in *Foxg1-Cre; Gnai2del/del; Gnaiflox/flox* mutants might preserve some functional GNAI3 proteins for a longer time and explain the apparent delay in OHC1–2 misorientation. To test this idea, we delayed GNAI downregulation by ptxA and asked whether this would result in a milder, delayed OHC1–2 misorientation phenotype as seen in *Gnai2; Gnai3* mutants. For that goal, we used the *Atoh1-Cre* driver which is only active in post-mitotic HC (*Matei et al., 2005*). Strikingly, while P0 OHC1 were inverted in *Atoh1-Cre; DIO-ptxA* as in *Foxg1-Cre; DIO-ptxA*, OHC2 generally pointed laterally (90°) in *Atoh1-Cre; DIO-ptxA* as in *Gnai2; Gnai3* mutants at the E17.5 base and the P0 apex (*Figure 8—figure supplement 1B*). However, a large proportion of OHC2 were inverted at the P0 base, supporting the hypothesis of a delayed inversion following delayed GNAI loss-of-function.

Together, these results first show that the *ptxA* misorientation pattern is present in *Gnai2; Gnai3* double mutants. Differences in severity between models can be explained by different timing of GNAI inactivation. Of note, the only surviving *Gnai2; Gnai3* double mutant we obtained suggests that OHC1–2 inversion is maintained in adults (*Figure 2A*), as in *ptxA* and *Gpr156* mutants (*Figure 2A*; *Kindt et al., 2021*). Endogenous GNAI proteins are thus integral for OHC1–2 to reverse and adopt a proper lateral orientation during development, and at least transiently to maintain this lateral orientation. Second, these results also suggest that the GNAI activities required for symmetry breaking, lateral HC orientation, and for hair bundle morphogenesis are qualitatively different (*Figure 9*). *Gnai2; Gnai3* mutants are more severely affected considering symmetry breaking and hair bundle morphogenesis, whereas *DIO-ptxA* mutants appear more severely affected considering OHC1–2 orientation. Of note, however, IHC and OHC3 were severely misoriented in *Gnai2; Gnai3* mutants at all stages and positions analyzed, but much less affected in *DIO-ptxA* mutants (*Figure 8*; *Figure 8—figure supplement 1A*). We discuss below how different GNAI protein identity, dose, timing, and upstream regulators may underlie different roles (*Figure 9*) and explain differences in phenotype across mutant models.

## Discussion

By examining a large collection of single and combined mutations in *Gnai/o* genes (*Gnai1*, *Gnai2*, *Gnai3*, *Gnao1*), we assign here three distinct roles for GNAI proteins during the apical polarization of a developing HC (*Figure 9*): (a) to drive the centrifugal migration of the basal body when a prospective HC breaks planar symmetry, (b) to orient this migration along the PCP axis in a binary manner, and (c) to position and elongate stereocilia during hair bundle morphogenesis. Key results in the study include demonstrating that endogenous GNAI proteins indeed serve early functions (a) and (b), since to date only hair bundle defects (c) were reported in knock-out mouse models where *Gnai* genes were targeted (*Beer-Hammer et al., 2018*; *Mauriac et al., 2017*). Previous studies suggested that GNAI proteins partner with different regulators to fulfill different roles, organizing and elongating stereocilia

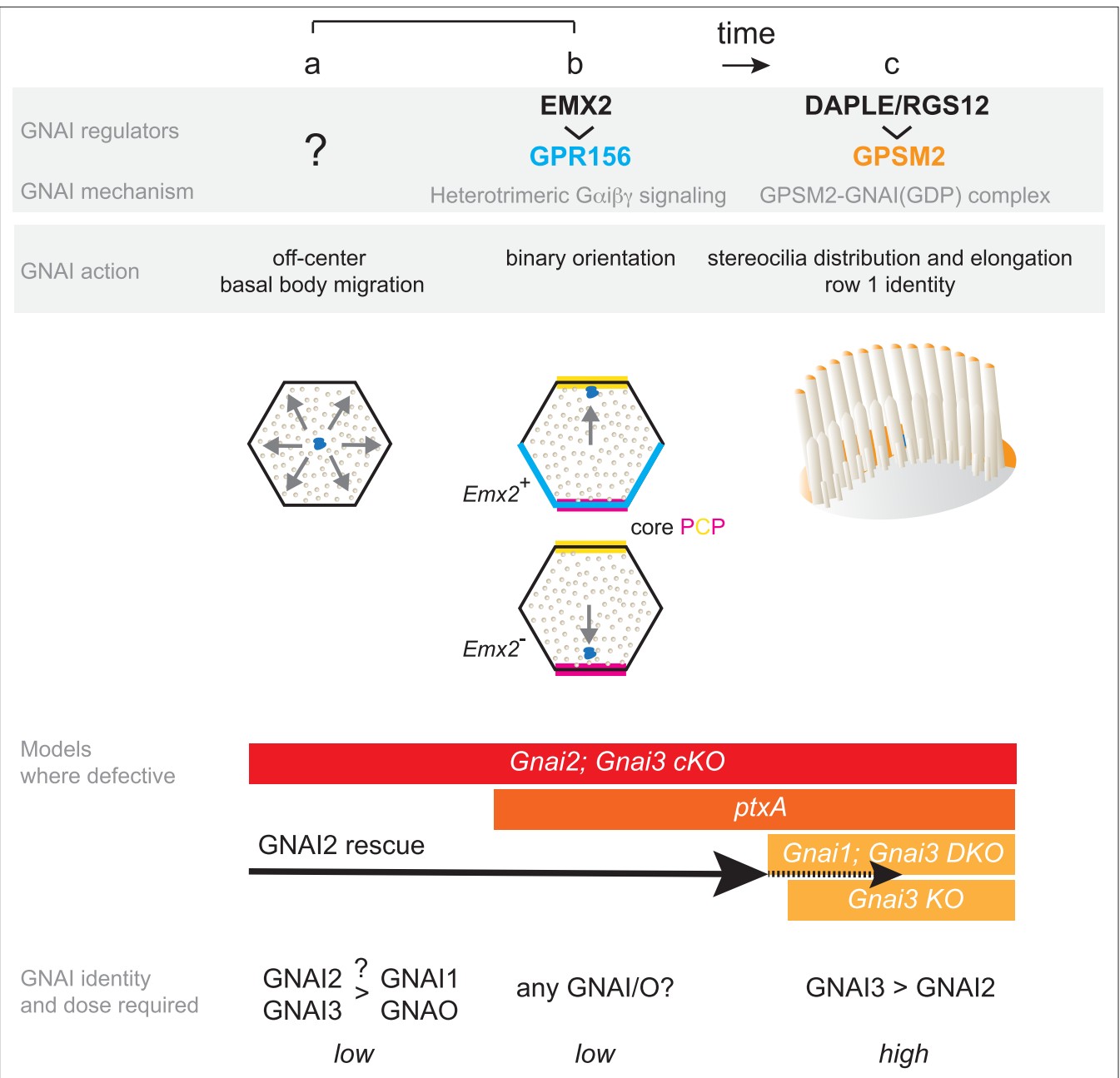

**Figure 9.** Summary and model: three roles validated in vivo for GNAI proteins during hair cell (HC) polarized morphogenesis. GNAI proteins are required for the off-center migration of the basal body, and thus HC symmetry breaking (**a**). This activity is distinct from defining binary HC orientation downstream of GPR156 (**b**), as *Gpr156* mutant HC have off-center basal bodies and normal hair bundles. Finally, GNAI partner with GPSM2 to shape the hair bundle and elongate stereocilia (**c**). The specific identity and the dosage of GNAI proteins required differ for each role. GNAI3 is the principal architect of hair bundle development (**c**), with GNAI2 playing an important but progressively waning role. High amounts of GNAI3/GNAI2 are required with GPSM2. A lower dose of GNAI/O proteins is required for embryonic role, and we speculate that GNAI2 and GNAI3 play a prominent role for symmetry breaking (**a**) whereas any GNAI/O protein may effectively signal downstream of GPR156 for proper HC orientation (**b**). A GNAI/O regulator for symmetry breaking remains to be identified.

by binding the scaffold GPSM2 (c), and reversing HC orientation in *Emx2*-expressing HC downstream of the receptor GPR156 (b).

## Different GNAI/O identity, dosage, and timing underlie different roles

Interestingly, besides involving different regulators, each role also appears to involve different dosage and specific identity of the underlying GNAI protein(s) at different times during HC development (*Figure 9*). For postnatal hair bundle morphogenesis (c), GNAI proteins are sequestered in the GDP-bound state by GPSM2, forming a complex highly enriched at the bare zone and at row 1 stereocilia tips that can be reliably immunodetected (*Akturk et al., 2022*; *Kindt et al., 2021*). In these two sub-cellular compartments, the identity of the GNAI proteins at work matters, with GNAI3 playing a required and prominent role while GNAI2 is important yet not required. Although one of our GNAI antibodies can detect GNAI1, we did not observe GNAI1 signals in HC (*Figure 4*), and *Gnai1* mutants did not exhibit significant hair bundle defects or ABR threshold shifts (*Figure 2*; *Figure 3A*). On the other hand, we observed a trend toward more severe stereocilia defects and more elevated ABR thresholds in *Gnai1; Gnai3* compared to single *Gnai3* mutants (*Figure 2C–E*; *Figure 3D*). Together, these results suggest that GNAI1 plays a minor, or no role in hair bundle development, with differences in the genetic background between strains possibly explaining the latter results. Similarly, we did not observe HC or auditory defects in absence of GNAO.

In stark contrast with its inability to rescue hair bundle morphogenesis (c), GNAI2 can fully rescue the off-center migration of the basal body (a) and its direction which defines HC orientation (b) when both GNAI1 and GNAI3 are constitutively absent. These two early roles are qualitatively different because they do not strictly reflect a difference in timing (a and b occur at the same time) or GNAI/O dose. Symmetry breaking (a) is affected in *Gnai2; Gnai3* double mutants but is intact in the *ptxA* model despite early (*Foxg1-Cre*) and high (caggs promoter) expression of untagged ptx. On the contrary, orientation of OHC1–2 (but not IHC and OHC3) is most severely affected in the *DIO-ptxA* model, with *Gnai2; Gnai3* mutants showing delayed defects. PtxA downregulates but does not abolish GNAI/O proteins, and we thus conclude that ptxA cannot achieve a loss of GNAI2 and GNAI3 function as extensive as in double targeted mutants. Among GNAI/O proteins, GNAI2 and GNAI3 may be more specifically required for HC to break symmetry, but other GNAI/O proteins and/or other factors may participate as well since a majority of HC eventually break symmetry even in *Gnai2; Gnai3* mutants. On the other hand, GPR156 is known to signal via heterotrimeric GNAI proteins (*Watkins and Orlandi, 2021*), and we speculate that any GNAI/O protein may equally relay signals to regulate HC orientation. PtxA would thus be a particularly effective way to disrupt the orientation role (b), with GNAI1 and/or GNAO partially rescuing orientation in *Gnai2; Gnai3* mutants. Our results also suggest that GNAI/O proteins are not only required for OHC1–2 to adopt a normal lateral instead of medial orientation upon symmetry breaking (normal GPR156-driven reversal; *Figure 9*), but also to maintain this lateral orientation, at least transiently. Delayed GNAI/O inactivation indeed appears to result in initially normal OHC1–2 lateral orientation followed by a switch to medial orientation in a proportion of cells (abnormal inversion; *Figure 8*; *Figure 8—figure supplement 1B*).

Regarding dosage, we propose that embryonic activities (a, b) run on low GNAI/O amounts compared to hair bundle functions (c). Of note, GNAI/O proteins are not immunodetected along with GPR156 in HC (b) but are easily detected with GPSM2 at the bare zone and stereocilia tips (*Akturk et al., 2022*; *Kindt et al., 2021*). This may be because GNAI/O proteins in heterotrimeric G protein complexes cycle dynamically between a GDP and GTP-bound state at low dose and escape immunodetection. In this context, it should be noted that a GNAI/O regulator for symmetry breaking and its mode of action remain unknown (*Figure 9*, role a).

A time and dosage dependence for early GNAI roles can explain why our conditional *Gnai2; Gnai3* double mutant model is more severely affected than a comparable conditional *Gnai2; Gnai3* model where only hair bundle defects (c) were reported (*Beer-Hammer et al., 2018*). While Beer-Hammer and colleagues used *Foxg1-Cre* as a driver as we did, combined GNAI2 and GNAI3 inactivation necessitated Cre recombination at four floxed loci (*Gnai2^flox/flox^*; *Gnai3^flox/flox^*). In contrast, our model necessitates recombination at two loci only (*Gnai2^del/del^*; *Gnai3^flox/flox^*). We thus conclude that our model achieves an earlier and further loss of GNAI2/GNAI3 activity. This hypothesis is supported by extensive postnatal lethality in our model, whereas Beer-Hammer and colleagues could analyze *Gnai2; Gnai3* double mutants as adults. Our work thus usefully reconciles previous reports about GNAI functions

in HC, notably showing that inactivating endogenous GNAI proteins does produce early defects (a, b) so far only observed with ptx. We thus validate all HC defects observed with ptx as physiologically relevant and specific to GNAI/O function.

## HC break of symmetry

Although cochlear HC expressing ptxA undergo normal symmetry breaking in vivo (this work and *Kindt et al., 2021*; *Tadenev et al., 2019*; *Tarchini et al., 2013*; *Tarchini et al., 2016*), a fraction of utricular and saccular HC expressing myc:ptxA showed an abnormally central basal body (*Kindt et al., 2021*). This suggests that a higher dose of GNAI/O is required for the off-center migration of the basal body in vestibular compared to cochlear HC, making vestibular HC more susceptible to ptxA. 'Ciliary' proteins are required for ciliogenesis, including kinocilium formation and maintenance, and also play non-ciliary functions (*May-Simera et al., 2015*; *Sipe and Lu, 2011*). Inactivation of intraflagellar transport protein IFT88 was reported to result in a central basal body and circular hair bundles in ~10% of OHC (*Jones et al., 2008*), but the underlying mechanism was not elucidated. A low proportion of symmetrical HC was also reported in absence of the CD2 isoform of the protocadherin PCDH15 that forms inter-stereocilia and kinocilium-stereocilia fibrous links during embryogenesis (*Webb et al., 2011*). It remains unclear whether GNAI/O activity participates in *Ift88* or *Pcdh15* mutant defects, or whether GNAI could be active at the basal body or the kinocilium.

## HC orientation

The PCP axis is defined by opposite asymmetric enrichment of the core PCP transmembrane proteins VANGL2 and FZD3/6 and their specific cytosolic partners at the apical junction of HC and support cells (*Deans, 2013*; *Montcouquiol and Kelley, 2020*; *Tarchini and Lu, 2019*). Previous work showed that regional *Emx2* expression reverses how the HC basal body migrates relative to uniform and early-set core PCP landmarks, notably establishing the line of polarity reversal in the utricle and saccule *Jiang et al., 2017*. EMX2 activates the GPR156 receptor by triggering its polarized enrichment at the apical HC junction, where downstream GNAI/O signaling appears to reverse how core PCP cues are interpreted and to repel the basal body (*Figure 9*, role b) (*Kindt et al., 2021*). EMX2 achieves this goal by preventing expression of the kinase STK32A that suppresses apical enrichment and polarization of the GPR156 protein (*Jia et al., 2023*). It is important to note that HC lacking GPR156 have a normal apical cytoskeleton, including a normally formed hair bundle, and only show orientation defects (*Kindt et al., 2021*). This indicates that the centrifugal migration of the basal body and the direction of this migration that defines early HC orientation are two distinct molecular mechanisms although both involve GNAI/O proteins (*Figure 9*; a *versus* b). Similarly, HC lacking GPSM2 are not inverted in orientation as in *Emx2*, *Gpr156*, *Gnai2*; *Gnai3*, or ptxA mutants (*Bhonker et al., 2016*; *Ezan et al., 2013*; *Tarchini et al., 2013*). This indicates that bare zone enrichment of the GPSM2-GNAI complex is a mechanism to shape and polarize the growth of the hair bundle, but not to migrate off-center or orient the basal body. GPSM2-GNAI is polarized on the side of the off-center basal body but occupies the HC apical membrane and not the apical junction where core PCP proteins regulate HC orientation (*Siletti et al., 2017*; *Tarchini et al., 2013*).

While both GPR156 (*Greene et al., 2023*; *Ramzan et al., 2023*) and GPSM2 (*Doherty et al., 2012*; *Walsh et al., 2010*) have been identified as human deafness genes, there is currently no evidence implicating a GNAI/O protein. This is most likely because all GNAI/O proteins, including GNAI3 which is more specifically required in HC for hair bundle morphogenesis (*Figure 9*), play ubiquitous and critical signaling roles in the context of heterotrimeric protein signaling across many cell types.

## Methods

### Mouse strains and husbandry

All mouse strains used in this work and strain-related information is summarized in *Supplementary file 1*. All primers used for genotyping are indicated in *Supplementary file 2* by strain.

### Strains from The Jackson Laboratory repository

*Gnai1^{neo}*; *Gnai3^{neo}* (*Gnai1^{tm1Lbi}*; *Gna3^{tm1Lbi}*; MGI: 5560183) (*Jiang et al., 2002*) carries a neo cassette in *Gnai1* exon 3 and a neo cassette replacing part of intron 5 and exon 6 in *Gnai3* in the 129S1/SvImJ

background. The single *Gnai1*neo and single *Gnai3*neo alleles were segregated from *Gnai1*neo; *Gnai3*neo by breeding with C57BL/6J mice and are consequently on a mixed 129S1/SvlmJ: C57BL/6J background. *Gnao1*neo (*Gnao1*tm1Lbi; MGI: 2152685) carries a neo cassette in *Gnao1* exon 6 in the 129S1/SvlmJ background (*Jiang et al., 2002*). The two *Cre* strains used in this work are *Atoh1-Cre* (*Tg(Atoh1-cre)1Bfri*; MGI: 3775845) expressing Cre in HC from E14.5 (*Matei et al., 2005*) and *Foxg1-Cre* (*Foxg1*tm1(cre)Skm; MGI: 1932522) expressing Cre in the prospective otic vesicle from E8.5 (*Hébert and McConnell, 2000*).

## Consortium strain

*Gnao1*flox (*Gnao1*tm1c(EUCOMM)Hmgu) was derived from the EUCOMM strain *Gnao1*tm1a(EUCOMM)Hmgu (MGI: 4456727; 'KO first, conditional ready'). The *Gnao1*flox conditional allele was produced via FLP-mediated recombination (FLP strain MGI: 4830363) to remove the FRT-flanked *LacZ-neo* cassette, leaving exon 3 floxed. *Gnao1*flox is on C57BL/6N background.

## Constitutive *Gnai2*del inactivation (see *Figure 2—figure supplement 1A*)

*Gnai2*del (*Gnai2*em1Btar; MGI: 6466534) was generated using delivery of CRISPR-Cas9 reagents in mouse zygotes via electroporation. The following guide RNAs (gRNAs) were used to delete exon 2, exon 3, and part of exon 4 in the C57BL/6J background. Upstream gRNA: CTGCCCTCTGTTCCAGGTGC, downstream gRNA: ATGCTTCCTGAAGACCTGTC. The resulting 681 bp deletion encompassed TGCTGGAG AGTCAGGGAAGA...CCTGAAGACCTGTCCGGTGT. The electroporation mixture consisted of the gRNAs with AltR- *Streptococcus pyogenes* Cas9 (SpCas9) V3 (Integrated DNA Technologies #1081059) in embryo-tested TE buffer (pH 7.5). Electroporation of zygotes was performed as described in *Qin et al., 2015*. In order to segregate away potential non-specific mutations, founders were bred for two generations with C57BL/6J animals to generate an N2 heterozygote stock. Because *Gnai2*del homozygotes showed low viability, this strain was used in a mixed C57BL/6J:FVB/J background that improved the proportion of homozygotes (see *Supplementary file 1*).

## Conditional *Gnai3*flox inactivation (see *Figure 2—figure supplement 1B*)

A plasmid-based donor vector was cloned to flank *Gnai3* exons 2 and 3 with *loxP* sites using the same CRISPR/Cas9 system as described above for *Gnai2*del. A fragment carrying a *loxP*, a genomic region including exons 2–3, the restriction site ClaI, a second *loxP*, and the restriction site XhoI was synthesized (Genscript) and cloned between 5' and 3' homology arms amplified by PCR using Gibson assembly. The following gRNAs were used and define the extremities of the floxed region: upstream, AGCTCACCAAAATTCCCATT, TAGGGGATATAGATCCAAAT, downstream, TTCCAGGACTCTGCAT GCGT, TACCGACGCATGCAGAGTCC. The gene editing reagents (gRNAs, donor vector, SpCas9) were microinjected in zygotes as described in *Qin et al., 2015*. To confirm insertion at the *Gnai3* locus, we performed long-range PCRs with a genomic primer located outside the homology arm on each side: *5' long-range:* F(external)_ TACTGAGATGAGAGACTGAGGG and R (internal)_TGGCTGAC ATCCTTTGATGGAC. The 3070 bp product was digested with XhoI, producing 2551+519 bp fragments upon donor insertion (flox allele). 3' long-range: F (internal)_TGAAAGGTAAAGGCAACGTG AG and R (external)_TGTGAGACAGGGTCTCTCTTTG. The 2966 bp product was digested with XhoI, producing 2000+966 bp fragments upon donor insertion (flox allele). In order to segregate away potential non-specific mutations, founders were bred for two generations with C57BL/6J animals to generate an N2 heterozygote stock.

## PtxA-expressing strains (see *Figure 2—figure supplement 1C and D*)

The *LSL-myc:ptxA* strain at the *Rosa26* locus was described previously (*Gt(ROSA)26Sor*em1(ptxA)Btar; MGI: 6163665) (*Tarchini et al., 2016*). The new *CAG-DIO-ptxA* strain line (see *Figure 2—figure supplement 1D*) was generated at the *Rosa26* locus using the Bxb1 attP(GT) integrase technology and *C57BL/6J-Gt(ROSA)26Sor*<em2Mvw>/Mvw (MGI: 6757188) as the host strain (*Low et al., 2022*). The donor vector consisted in a CAG promoter followed by the flipped ptxA coding sequence flanked by double inverted lox sites (loxP and lox2272; *Figure 2—figure supplement 1D*) and a bGHpA sequence. In order to exclude the prokaryotic vector backbone, the donor vector was prepared as a minicircle using the MC-Easy Minicircle Production kit (SystemBio, MN920A-1). Briefly, the donor plasmid insert

was cloned into a Minicircle Cloning Vector using PhiC31 integrase before transformation into the ZYCY10P3S2T *Escherichia coli* minicircle producer strain, which after induction with arabinose, results in the generation of the donor minicircle. To eliminate parental plasmid contamination, a restriction digest was performed, followed by MC-safe DNase treatment. The minicircle DNA was then purified by phenol-chloroform extraction and reconstituted in microinjection buffer (10 mM Tris; 0.1 mM EDTA pH 7.5). The *CAG-DIO-ptxA* strain was generated by pronuclear microinjection of the Bxb1 Integration reagents directly into zygotes of the host strain. These reagents included the Bxb1 mRNA (Trilink) at 100 ng/µl, RNasin (Promega) at 0.2 U/µl, and the donor minicircle DNA at 10 ng/µl combined in microinjection buffer. To confirm successful integration of the 3212 bp transgene, as well as identify random transgenics, the initial screening was performed using a four-PCR strategy. The In/Out Left and Right (IOL, IOR) PCRs, which bridge the recombined attachment sites, each contain one primer specific to the *Rosa26* locus and a second primer designed against the inserted sequence. The Transgene (TG) PCR amplifies a non-genomic sequence in the insert, and the Off-Target Integration (OTI) PCR was designed to detect non-recombined minicircle integration, presumably outside of the *Rosa26* locus.

IOL PCR (F1/R1), F1: GTCGCTCTGAGTTGTTATCAGT, R1: GCCAAGTAGGAAAGTCCCATAA (720 bp, WT = no band). IOR PCR (F2/R2), F2: GGTGATGCCGTTGTGATAGA, R2: TGTGGGAAGTCT TGTCCCTCCAAT (1013 bp, WT = no band). TG PCR (F2/R3) F2: 5'-GGTGATGCCGTTGTGATAGA, R3: CCACTTCATCGGCTACATCTAC (214 bp, WT = no band). OTI PCR (F3/R1) F3: GGGAGGATTGGG AAGACAATAG, R1: GCCAAGTAGGAAAGTCCCATAA (578 bp, WT = no band). 33 mice were born following 6 transfers totaling 117 injected embryos (28% survival), and 1/33 (3%) was identified as a founder and crossed to C57BL6/J to generate N1 offspring. The In/Out PCR confirmed that a copy of the transgene was inserted at the *Rosa26* locus, but the OTI strategy also gave a product, even after breeding for multiple generations. As breeding would have segregated a separate, random integration of the transgene, we performed Nanopore-based Cas9-targeted sequencing at the *Rosa26* locus (*Gilpatrick et al., 2020*; *Low et al., 2022*). This revealed that, as seen in some cases previously (*Low et al., 2022*), tandem insertion had occurred in this founder. Specifically, three consecutive copies of the *CAG-DIO-ptxA* transgene were inserted at the *Rosa26* locus in this strain. This did not impact specific Cre-based expression of ptxA because phenotypes observed in this new strain were comparable to the previous *LSL-myc:ptxA* strain (*Figure 6—figure supplement 1B–D*).

Experimental animals in the study ranged in age between E17.5 and P29 as indicated in each figure. Male and females were systematically included but sex was not tracked except for Auditory Brainstem Recordings because there is no evidence that sex influences HC orientation or hair bundle morphogenesis. Animals were maintained under standard housing conditions (14 hr light/10 hr dark cycle, ambient temperature, and normal humidity). All animal work was reviewed for compliance and approved by the Animal Care and Use Committee of The Jackson Laboratory (Animal Use Summary AUS #14012).

## Scanning electron microscopy

Temporal bones were isolated, punctured at the cochlear apex, and fixed by immersion for at least one overnight at 4°C in 2.5% glutaraldehyde (Electron Microscopy Sciences; 16200) and 4% paraformaldehyde (PFA, Electron Microscopy Sciences; 15710) in 1 mM $MgCl_2$, 0.1 M sodium cacodylate, 20 mM $CaCl_2$. Samples were rinsed and decalcified overnight in 4% EDTA. The auditory epithelium was then dissected into three pieces (cochlear base, mid, and apex) before progressive dehydration in an ethanol series (30–50–70–80–90–100%, at least 20 min per step) and chemical drying with hexamethyldisilazane (Electron Microscopy Sciences 50-243-18). Dry samples were mounted on aluminum stubs using double-sided carbon tape and sputter-coated with gold-palladium before imaging on a Hitachi 3000N VP electronic microscope at 20 kV.

## Immunofluorescence and antibodies

For embryonic and postnatal stages, temporal bones were immediately dissected to expose the sensory epithelium and fixed in PFA (4%; Electron Microscopy Sciences; 15710) for 1 hr at 4°C. After fixation, the tectorial membrane was removed, and samples were permeabilized and blocked in PBS with 0.5% Triton X-100 and bovine serum albumin (1%) for at least at 1 hr at room temperature. For adult stages, temporal bones were isolated and punctured at the cochlear apex to facilitate access

of the fixative. Samples were then immersion-fixed in PFA 4% for 1 hr at 4°C, rinsed in PBS, and incubated overnight in 4% EDTA for decalcification. Cochleae were next dissected in three pieces (cochlear base, mid, and apex), before permeabilization and blocking as described above. Primary and secondary antibodies were incubated overnight at 4°C in PBS with 0.025% sodium azide. Fluorescent dye-conjugated phalloidin was added to secondary antibodies. Samples were washed three times in PBS+0.05% Triton X-100 after each antibody incubation and post-fixed in PFA 4% for at least 1 hr at room temperature. Samples were then mounted flat on microscopy slides (Denville M102) using Mowiol as mounting medium (Calbiochem/MilliporeSigma 4759041), either directly under a 18×18 mm² #1.5 coverglass (VWR 48366-045) (postnatal cochleae) or using one layer of office tape as a spacer (adult cochleae). Mowiol (10% wt/vol) was prepared in 25% (wt/vol) glycerol and 0.1 M Tris-Cl pH 8.5. Primary antibodies used were:

> Rabbit anti-GNAI2, pt"GNAI2" (Proteintech, 11136-1-AP); the antigen is the full human GNAI2 protein
> Rabbit anti-GNAI3, scbt"GNAI3" (Santa Cruz Biotechnology, sc-262); the antigen is undisclosed but corresponds to a C-terminal region of the rat GNAI3 protein
> Rabbit anti-GNAO (Proteintech, 12635-1-AP)
> Mouse anti-acetylated alpha tubulin (Santa Cruz Biotechnology scbt-23950)
> Rabbit anti-GPSM2 (Sigma, A41537)
> Goat anti-GPSM2 (Thermo Fisher, PA5-18646)
> Rabbit anti-Pericentrin/PCNT (Biolegend/Covance, PRB-432C)
> Rat anti-ZO1 (Developmental Studies Hybridoma Bank, R26.4C)

Secondary antibodies from Thermo Fisher Scientific were raised in donkey and conjugated to Alexa Fluor (AF) 488, 555, or 647 (donkey anti-rabbit AF555 [A-31572], AF647 [A-31573], donkey anti-mouse AF647 [A-31571], donkey anti-rat AF488 [A-21208], donkey anti-goat AF555 [A-21432], AF647 [A-21447]). Fluorescent-conjugated phalloidins used to reveal F-actin were from Thermo Fisher Scientific (AF488 [A12379]; AF555 [A34005]) and Sigma-Aldrich (FITC [P5282]).

## Inner ear electroporation and cochlear culture

Inner ears from wild-type animals (mixed C57BL/6J-FVB/NJ background) were harvested at E13.5 in HBSS/5 mM HEPES. Caggs plasmid vectors (CMV early enhancer, beta-actin promoter) driving either mouse *Gnai1*, *Gnai2*, or *Gnai3* cDNA expression were mixed with Fast Green FCF (0.05%; Sigma, F7252) and injected at 2.5 mg/ml in the cochlear duct using a Wiretroll capillary and plunger (Drummond, 53507-426). Injected inner ears were next electroporated (BTX ECM830; 27 V, 27 ms, 6 square pulses at 950 ms intervals), the condensed mesenchyme was dissected away and the soft cochlear labyrinth was embedded in 50% Matrigel (Corning, CB40234). Cochlear explants were cultured for 6 days in DMEM with 10% fetal bovine serum and 10 μg/ml ciprofloxacin (Sigma, 17850). Explants were then fixed in 4% PFA for 15 min at room temperature before being processed for immunolabeling.

## Sample cohorts, image acquisition, and analysis

All quantifications include at least three animals per genotype. All graphs or their legends indicate the animal cohort size (N) as well as the number of HC or stereocilia (n) analyzed. When an experimental outcome was not quantified, at least three mutant and three control littermates encompassing two or more litters were analyzed, and figure panels illustrate a representative outcome observed in all samples of the same genotype. A single exception is adult *Foxg1-Cre; Gnai2*$^{del/del}$*; Gnai3*$^{flox/flox}$ data in *Figure 2A–F* where a single mutant animal was analyzed since we failed to obtain others at 3 weeks of age in spite of extensive breeding (see *Supplementary file 1*).

Confocal images were captured with an LSM800 line scanning confocal microscope, a 63× NA1.4 oil objective, the Airyscan detector in confocal mode (except in *Figure 5—figure supplement 1D and E* where the Airyscan detector was used in Airyscan mode), and the Zen 2.3 or Zen 2.6 software (Carl Zeiss AG). Raw Airyscan images in *Figure 5—figure supplement 1D and E* were processed in Zen 2.6 selecting for automatic strength. Unless stated otherwise in the legends, images show a single optical z plane. To quantify HC eccentricity (*Figure 7*), images were captured with a Leica DM5500B widefield microscope, a 63× oil objective, a Hamamatsu ORCA-Flash4.0 sCMOS camera, and the Leica Application Suite (LasX) software (Leica Microsystems). All confocal images in the same experiment

were acquired using the same laser intensity and gain, and were then processed in Adobe Photoshop (CC2020) where the same image treatment was applied across conditions.

To measure stereocilia length and width in IHC (*Figure 2C and D*), SEM samples were imaged laterally at ×10,000 magnification and with an appropriate tilt (from 0° to 30°) to bring stereocilia parallel to the imaging plane and minimize parallax. To quantify the number of rows in adult IHC and the number of stereocilia in their first row (*Figure 2E and F*), SEM samples were imaged medially at ×5000 magnification. To measure the length of OHC hair bundle wings (*Figure 2G*, *Figure 2—figure supplement 1G*), OHC were imaged top down at ×5000. Stereocilia width was measure at half-length and all measurements were done with the straight-line tool in Fiji.

To quantify GNAI signal intensity in *Gnai1*$^{neo}$; *Gnai3*$^{neo}$ mutants (*Figure 5C*, *Figure 5—figure supplement 1A*), Z-stack series were acquired at the mid cochlear position. A single Z-slice was chosen at the bare zone level, another one at the stereocilia tip level, each based on strongest signal in that compartment. The vertex of the V-shaped hair bundle was used to divide the HC apical surface in two equal halves. In each half (left and right), regions of interest (ROIs) were drawn using the polygon selection tool in Fiji to encompass all signals in that apical compartment, and mean gray values were measured. For each image, background signal was measured and averaged, and subtracted from all measurements in the same image.

To quantify surface area of the bare zone or surface area in half-OHC as well as the length of half hair bundles (*Figure 5G*, *Figure 5—figure supplement 1C*; *Figure 6D and H*), Z-stack series were acquired at the cochlear base and a single Z-slice was selected at apical junction level using ZO1 (*Figure 5G*, *Figure 5—figure supplement 1C*) or F-actin (*Figure 6D and H*) as reference. To measure apical surface area and bundle length in half-OHC, the PCNT-stained basal body was used to divide the apical surface in two halves (*Figure 5G*, *Figure 5—figure supplement 1C*). The ZO1-positive cell outline along with the dividing line at the basal body level were used as reference to measure surface area with the polygon selection tool in Fiji. To quantify the bare zone surface area (*Figure 6D and H*), a ROI was drawn around the total apical surface lacking phalloidin (F-actin) signals. The length of the F-actin-stained hair bundle was measured using the straight-line tool in Fiji.

To determine HC eccentricity (*Figure 7*), the geometrical center of the HC apical surface was determined as the intersection of two orthogonal lines representing the maximal diameter of the cell along the cochlear longitudinal and radial axes. A vector (BB) was drawn from the center to the PCNT-labeled basal body, and eccentricity was calculated as the ratio of BB length over the cell radius (r) along the same trajectory using F-actin-labeled apical junction as landmark (see *Figure 7*). To determine cell orientation (*Figure 8* and *Figure 8—figure supplements 1 and 2*), the angle (α) separating the longitudinal axis of the organ of Corti from the BB vector was measured with the angle tool in Fiji (see *Figure 8*). Both right and left cochleae were used and angles were measured so that 0° pointed toward the cochlear base and 90° toward the cochlear periphery (lateral). Eccentricity and angles were measured at the cochlear positions indicated (base ~20%, mid ~50%, apex ~80% of the cochlear length starting from the base).

## ABR tests

All tests were performed in a sound-attenuating chamber, and body temperature of the anesthetized animals was maintained at 37°C using a heating pad (FHC Inc). Animals from all strains except *Atoh1-Cre; Gnao1*$^{flox}$ (*Figure 3—figure supplement 1B*, see below) were anesthetized with a mix of ketamine and xylazine (1 mg and 0.8 mg per 10 g of body weight, respectively) and tested using the RZ6 Multi-I/O Processor System coupled to the RA4PA 4-channel Medusa Amplifier (Tucker-Davis Technologies). ABRs were recorded after binaural stimulation in an open field by tone bursts at 8, 16, 32, and in some cases 40 kHz generated at 21 stimuli/s, and a waveform for each frequency/dB level was produced by averaging the responses from 512 stimuli. Subdermal needles were used as electrodes, the active electrode inserted at the cranial vertex, the reference electrode under the left ear, and the ground electrode at the right thigh.

*Atoh1-Cre; Gnao1*$^{flox}$ animals (*Figure 3—figure supplement 1B*) were anesthetized with tribromoethanol (2.5 mg per 10 g of body weight) and tested with the Smart EP evoked potential system from Intelligent Hearing Systems (IHS). ABRs were recorded after single ear stimulation (right ear), using ear tubes speakers (ER3C insert earphones, IHS) delivering tone bursts at 8, 16, and 32 kHz generated at 40 stimuli/s. Electrodes were positioned as previously described except for the ground

electrode that was placed at the base of the tail. ABR thresholds were obtained for each frequency by reducing the SPL by 5 decibels (dB) between 90 and 20 dB to identify the lowest level at which an ABR waveform could be recognized. We compared waveforms by simultaneously displaying 3 or more dB levels on screen at the same time.

## Statistical analyses

All data were plotted in Prism 9 (GraphPad), except for circular diagrams of HC orientation (*Figure 8*; *Figure 8—figure supplements 1 and 2*) that were generated with R (4.2.2) and Rstudio (2022.12.0+353).

All data except for angles in circular diagrams (*Figure 8* and *Figure 8—figure supplements 1 and 2*) were plotted individually. Distribution was framed with 25–75% whisker boxes where exterior lines show the minimum and maximum, the middle line represents the median, and + represents the mean. Potential differences in data distribution between genotypes were tested for significance using nested (hierarchical) t-test except for ABR thresholds (*Figure 3* and *Figure 3—figure supplement 1*) where a two-way ANOVA with Sidak's multiple comparison post hoc test was used. Nested t-tests help avoid pseudoreplication by taking into consideration the data structure, here specifically variance in each animal (*Eisner, 2021*; *Krey et al., 2023*). OHC half-bundle lengths (*Figure 2G*, *Figure 2—figure supplement 1G*) were plotted as paired left and right values in the same HC and a potential difference in data variance between genotypes was tested using an F-test. GNAI signal intensity at the OHC bare zone and stereocilia tips (*Figure 5C*, *Figure 5—figure supplement 1A*) as well as OHC surface area and half-bundle length (*Figure 5G*, *Figure 5—figure supplement 1C*) were plotted and a simple linear regression curve was calculated and drawn for each pair of datasets. A correlation between variables was addressed with Pearson correlation test. Exact p-values were indicated on each graph when non-significant (p>0.05) and otherwise summarized as follows: p<0.0001****, p<0.001***, p<0.01**, p<0.05*.

Angle frequency distribution (*Figure 8* and *Figure 8—figure supplements 1 and 2*) was plotted in circular diagrams using the R package dyplr and the coord_polar function of the ggplot2 package to organize data and produce the graphs, respectively. The angle formed by the red line indicates the circular mean and the length of the arc at the end of the red line indicates the mean circular deviation. Both values were obtained using the colstats function in the R circular package. All scripts used in this work are posted on GitHub.

## Acknowledgements

We are grateful to Elli Hartig for reading and commenting on the manuscript. We thank Simon Lesbirel for nanopore-based Cas9-targeted sequencing of the *DIO-ptxA* strain. We are grateful to The Jackson Laboratory Genome Engineering Technology and Reproductive Science services for their help with generating the *Gnai2*del and *Gnai3*flox strains, and cryorecovering the *Gnai1*neo; *Gnai3*neo and *Gnao1*neo strains, respectively. AJ was supported by a postdoctoral fellowship from Fondation Pour l'Audition (2018–2020; FPA RD-2018-3). This work was supported by the National Institute on Deafness and Other Communication Disorders grants R01DC015242 and R01DC018304 (to BT).

## Additional information

### Funding

| Funder | Grant reference number | Author |
| --- | --- | --- |
| National Institute on Deafness and Other Communication Disorders | R01 DC015242 | Basile Tarchini |
| National Institute on Deafness and Other Communication Disorders | R01 DC018304 | Basile Tarchini |
| Fondation Pour l'Audition | FPA RD-2018-3 | Amandine Jarysta |

| Funder | Grant reference number | Author |
|--------|------------------------|--------|

The funders had no role in study design, data collection and interpretation, or the decision to submit the work for publication.

## Author contributions

Amandine Jarysta, Data curation, Formal analysis, Validation, Investigation, Visualization, Methodology, Writing - original draft, Writing – review and editing; Abigail LD Tadenev, Data curation, Formal analysis, Validation, Investigation, Visualization, Methodology, Writing – review and editing; Matthew Day, Investigation; Barry Krawchuk, Resources, Software; Benjamin E Low, Michael V Wiles, Resources; Basile Tarchini, Conceptualization, Data curation, Formal analysis, Supervision, Funding acquisition, Validation, Investigation, Visualization, Methodology, Writing - original draft, Project administration, Writing – review and editing

## Author ORCIDs

Amandine Jarysta (ID) http://orcid.org/0000-0002-9519-3559
Matthew Day (ID) http://orcid.org/0000-0002-0422-2132
Basile Tarchini (ID) http://orcid.org/0000-0003-2708-6273

## Ethics

All animal work was reviewed for compliance and approved by the Animal Care and Use Committee of The Jackson Laboratory (Animal Use Summary AUS #14012).

Reviewer #1 (Public Review): https://doi.org/10.7554/eLife.88186.3.sa1
Reviewer #2 (Public Review): https://doi.org/10.7554/eLife.88186.3.sa2
Author response https://doi.org/10.7554/eLife.88186.3.sa3

# Additional files

## Supplementary files

• Supplementary file 1. Table providing mouse strain details.
• Supplementary file 2. Table providing genotyping strategies.
• MDAR checklist

## Data availability

The research data that support the findings in this study, including detailed cohort sizes, graphed values and statistical analysis, are available in Zenodo with the identifier DOI: https://doi.org/10.5281/zenodo.10790739. The R code to produce circular diagrams representing hair cell orientation is available in GitHub at https://github.com/Tarchini-Lab/R-code-for-circular-diagrams, (copy archived at *Tarchini-Lab, 2024*).

The following dataset was generated:

| Author(s) | Year | Dataset title | Dataset URL | Database and Identifier |
|-----------|------|---------------|-------------|-------------------------|
| Jarysta A, Tarchini B | 2024 | Inhibitory G proteins play multiple roles to polarize sensory hair cell morphogenesis | https://doi.org/10.5281/zenodo.10790739 | Zenodo, 10.5281/zenodo.10790739 |

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
