## [Editor Report · eLife assessment]

This study examines an **important** aspect of the development of the auditory system, the role of guanine nucleotide-binding protein subunits, GNAIs, in stereociliary bundle formation and orientation, by examining bundle phenotypes in multiple compound GNAI mutants. The experiments are highly rigorous and thorough and include detailed quantifications of bundle morphologies and changes. The depth and care of the study are impressive, with **convincing** results regarding the roles of GNAIs in stereociliary bundle development. Further, the reviewers believe this to be the definitive study of the role of GNAIs in bundle orientation and development.

---

## [Referee Report · Reviewer #1 (Public Review)]

A subclass of inhibitory heterotrimeric guanine nucleotide-binding protein subunits, GNAI, has been implicated in sensory hair cell formation, namely the establishment of hair bundle (stereocilia) orientation and staircase formation. However, the former role of hair bundle orientation has only been demonstrated in mutants expressing pertussis toxin, which blocks all GNAI subunits, but not in mutants with a single knockout of any of the Gnai genes, suggesting that there is a redundancy among various GNAI proteins in this role. Using various conditional mutants, the authors concluded that GNAI3 is the primary GNAI proteins required for hair bundle morphogenesis, whereas hair bundle orientation requires both GNAI2 and GNAI3.

Strength

Various compound mutants were generated to decipher the contribution of individual GNAI1, GNAI2, GNAI3 and GNAIO in the establishment of hair bundle orientation and morphogenesis. The study is thorough with detailed quantification of hair bundle orientation and morphogenesis, as well as auditory functions.

The revised manuscript has clarified the phenotypic differences raised between the Gnai2/3 double mutants and Ptx mutant phenotypes and resolved the weakness pointed out in the previous submission. These results further illustrate the dynamic requirement of Gnai/O in hair bundle establishment and is an important contribution to the field.

---

## [Referee Report · Reviewer #2 (Public Review)]

Jarysta and colleagues set out to define how similar GNAI/O family members contribute to the shape and orientation of stereocilia bundles on auditory hair cells. Previous work demonstrated that loss of particular GNAI proteins, or inhibition of GNAIs by pertussis toxin, caused several defects in hair bundle morphogenesis, but open questions remained which the authors sought to address. Some of these questions include whether all phenotypes resulting from expression of pertussis toxin stemmed from GNAI inhibition; which GNAI family members are most critical for directing bundle development; whether GNAI proteins are needed for basal body movements that contribute to bundle patterning. These questions are important for understanding how tissue is patterned in response to planar cell polarity cues.

To address questions related to the GNAI family in auditory hair cell development, the authors assembled an impressive and nearly comprehensive collection of mouse models. This approach allowed for each Gnai and Gnao gene to be knocked out individually or in combination with each other. Notably, a new floxed allele was generated for Gnai3 because loss of this gene in combination with Gnai2 deletion was known to be embryonic lethal. Besides these lines, a new knockin mouse was made to conditionally express untagged pertussis toxin following cre induction from a strong promoter. The breadth and complexity involved in generating and collecting these strains makes this study unique, and likely the authoritative last word on which GNAI proteins are needed for which aspect of auditory hair bundle development.

Appropriate methods were employed by the authors to characterize auditory hair bundle morphology in each mouse line. Conclusions were carefully drawn from the data and largely based on excellent quantitative analysis. The main conclusions are that GNAI3 has the largest effect on hair bundle development. GNAI2 can compensate for GNAI3 loss in early development but incompletely in late development. The Gnai2 Gnai3 double mutant recapitulates nearly all the phenotypic effects associated with pertussis toxin expression and also reveals a role for GNAIs in early movement of the basal body. This comprehensive study builds on earlier reports, both uncovering new functions and putting previously putative functions on solid ground.

---

## [Author Response]

The following is the authors’ response to the current reviews.

**Recommendations for the authors:**

**Reviewer #1 (Recommendations For The Authors):**
Hats off to the authors for taking time to decipher the seemingly subtle but important differences between the Gnai2/3 double mutant and Ptx mutant phenotypes. These results further illustrate the dynamic requirement of Gnai/0 in hair bundle establishment. I have some minor suggestions for the authors to consider and it is up to the authors to decide whether to incorporate them:

We decided to make the current (revised) version the version of record, and we explain why below. Please include these comments in the review+rebuttal material.

(1) The abstract could be modified to reflect the revised interpretations of the results.

Response: the abstract is high-level and the changes in interpretation in the revised manuscript do not modify the message there. Briefly, the abstract only states that Gnai2; Gnai3 double mutants recapitulate two defects previously only observed with pertussis toxin. There is no claim about the timing or dose of GNAI proteins involved.

(2) The three rows of OHCs are like a different beast from each other. Mireille Montcouquiol's lab has demonstrated that there is a differential requirement for Gnai3 in hair bundle orientation among the three rows of OHCs. The results described in this manuscript support this notion as well.

To clarify, Gnai3 inactivation does not affect OHC orientation. Only pertussis toxin, and in this work Gnai2; Gnai3 double mutants, do. The Montcouquiol lab showed different degree of OHC1, OHC2 and OHC3 misorientation upon use of pertussis toxin in vitro using cochlear explants (Ezan et al 2013). We showed the same thing in vivo using transgenic models (Tarchini et al 2013; Tarchini et al 2016). The different OHC responses by row and corresponding citations are mentioned in several locations in the manuscript, including first on line 112 in the Introduction and in Fig. 1C in a graphical summary.

(3) I wonder if "compensate" or "redundancy" may be a better term to use than "rescue" in the Discussion and figure.

Use of “rescue” in the Discussion is line 603 and 604. We think that “rescue” is appropriate to refer to the ability of GNAI2 to compensate for the loss of GNAI1 and GNAI3 in mutant context. We would argue that these different wordings are largely interchangeable and do not change the message.

**Author Response**

The following is the authors’ response to the original reviews.

We really appreciate the time the reviewers spent reading and commenting on the original manuscript. Although they were positive already, we decided to spend some time to address the main comments with new experiments as thoroughly as possible in a new manuscript version. We also heavily edited some sections accordingly: (1) we delayed pertussis toxin activation in hair cells with Atoh1-Cre to show that the resulting misorientation phenotype is delayed compared to FoxG1-Cre results, as also seen in Gnai2; Gnai3 double mutants. It follows that Gnai2; Gnai3 and pertussis mutants do share a similar misorientation profile, and that GNAI proteins are required to normally reverse OHC1-2 (from medial to lateral), but also to maintain the lateral orientation, at least transiently. (2) We experimentally verified that one of our GNAI antibodies can indeed detect GNAI1, and consequently that absence of signal in Gnai2; Gnai3 double mutants is evidence that GNAI1 is not involved in apical hair cell polarization. We believe these changes strengthen the manuscript and its conclusions.

**Reviewer #1 (Public Review):**
A subclass of inhibitory heterotrimeric guanine nucleotide-binding protein subunits, GNAI, has been implicated in sensory hair cell formation, namely the establishment of hair bundle (stereocilia) orientation and staircase formation. However, the former role of hair bundle orientation has only been demonstrated in mutants expressing pertussis toxin, which blocks all GNAI subunits, but not in mutants with a single knockout of any of the Gnai genes, suggesting that there is a redundancy among various GNAI proteins in this role. Using various conditional mutants, the authors concluded that GNAI3 is the primary GNAI proteins required for hair bundle morphogenesis, whereas hair bundle orientation requires both GNAI2 and GNAI3.StrengthVarious compound mutants were generated to decipher the contribution of individual GNAI1, GNAI2, GNAI3 and GNAIO in the establishment of hair bundle orientation and morphogenesis. The study is thorough with detailed quantification of hair bundle orientation and morphogenesis, as well as auditory functions.WeaknessWhile the hair bundle orientation phenotype in the Foxg1-cre; Gnai2-/-; Gnai3 lox/lox (double mutants) appear more severe than those observed in Ptx cKO mutants, it may be an oversimplification to attribute the differences to more GNAI function in the Ptx cko mutants. The phenotypes between the double mutants and Ptx cko mutants appear qualitatively different. For example, assuming the milder phenotypes in the Ptx cKO is due to incomplete loss of GNAI function, one would expect the Ptx phenotype would be reproducible by some combination of compound mutants among various Gnai genes. Such information was not provided. Furthermore, of all the double mutant specimens analyzed for hair bundle orientation (Fig. 8), the hair bundle/kinocilium position started out normally in the lateral quadrant at E17.5 but failed to be maintained by P0. This does not appear to be the case for Ptx cKO, in which all affected hair cells showed inverted orientation by E17.5. It is not clear whether this is the end-stage of bundle orientation in Ptx cKO, and the kinocilium position started out normal, similar to the double mutants before the age of analysis at E17.5. Understanding these differences may reveal specific requirements of individual GNAI subunits or other factors are being affected in the Ptx mutants.

This criticism was very useful and prompted new experiments as well as a change in data presentation and a fundamental rewrite regarding hair cell orientation. These changes are detailed below. Of note, however, please let us clarify that the original manuscript did show that the ptxA orientation phenotype is reproduced to some extent in Gnai2; Gnai3 double mutants (previously Fig. 8 and corresponding text line 505). We showed that OHC1-2 are also inverted in the double mutant, although at a later differentiation stage. We recognize that similarities in hair cell misorientation between ptxA and Gnai2; Gnai3 DKO were not explained and discussed well enough. This part of the manuscript has been re-worked extensively, and we hope that along with new results, comparisons between mutant models are easier to follow and understand. We notably fully adopted the idea that there are qualitative differences between ptxA and Gnai2; Gnai3 mutants, and not only a difference in the remaining “dose” of GNAI activity.

**Recommendations for the authors:**

**Reviewer #1 (Recommendations For The Authors):**
Comments related to clarification of the weakness:(1) In general, hair bundle orientation in the double mutants is established in the lateral quadrant of the cochlea before being inverted (Fig. 8). These results are intriguing because the lateral orientation is the correct position for these hair bundles normally and Gnai proteins are thought to be required to get the kinocilium to the lateral position. This process appears to proceed normally in the double mutants but the kinocilium reverted to the medial default position over time, which suggests that Gnai2 and Gnai3 are only required for the maintenance and not the establishment of the kinocilium in the lateral position. Is this phenotype qualitatively similar in the Ptx cKO?

We addressed these issues with two types of modifications to the data:

(1) We modified the eccentricity threshold used at E17.5 in Fig. 8 (orientation) to be more stringent, using 0.4 (instead of 0.25 previously) in both controls and mutants. This means that we now only graph the orientation of cells where eccentricity is more marked. The rationale is that at early stages, it is challenging to distinguish immature vs defective near-symmetrical cells. We kept a threshold of 0.25 at P0 when the hair cell apical surface is larger and better differentiated (Fig. 8C-D). Importantly, the dataset remains rigorously identical. This change usefully highlights that a large proportion of OHC1 is in fact inverted (oriented medially) at E17.5 in Gnai2; Gnai3 double mutants at the cochlear mid, as also seen in the ptxA model at the same stage and position (see new Fig. 8A). At the E17.5 base (Fig. 8B), a slightly more mature position, the outcome is unchanged (the majority of OHC1 are inverted using either a 0.25 or 0.4 threshold in double mutants and in ptxA).

Interestingly however, the orientation trend is unchanged for OHC2: OHC2 remain oriented largely laterally (i.e. normally) at the E17.5 mid and base in Gnai2; Gnai3 double mutants even with a raised eccentricity thresholds, whereas by contrast OHC2 in ptxA are inverted at these stage and positions. In the double mutant, OHC2 only become inverted at the P0 base (Fig. 8D). This suggests that there are similarities (OHC1) but also differences (OHC2s) between the two mouse models, and that double mutants show a delay in adopting an inverted orientation compared to ptxA. Of note, OHC2 have been shown to differentiate later than OHC1 (for example, Anniko 1983 PMID:6869851).

(2) To directly test the idea that the misorientation phenotype (inverted OHC1-2) is comparable between the two models but delayed in Gnai2; Gnai3 mutants, we performed a new experiment and added new results in the manuscript. We delayed ptxA action by using Atoh1-Cre (postmitotic hair cells) instead of FoxG1-Cre (otic progenitors). Remarkably, this produced a pattern of OHC1-2 misorientation more similar to Gnai2; Gnai3 mutants: at the E17.5 base and P0 apex, OHC2 were still largely oriented laterally (normally) in Atoh1-Cre; ptxA as in Gnai2; Gnai3 mutants whereas at the P0 base a large proportion of OHC2 were inverted (Fig. 8 Supp 1B). OHC1 were inverted at all stages and positions in the Atoh1-Cre as in the FoxG1-Cre; ptxA model. For Atoh1-Cre; ptxA, we only illustrated OHC1 and OHC2 and did not add E17.5 mid or P0 mid results because other cell types and stage/positions did not provide additional insight. In addition, we are well aware that the full FoxG1-Cre; ptxA and Gnai2; Gnai3 results for 4 cells types (IHC, OHC1-3) and 5 stages/positions is already a lot of data for cell orientation.

These results suggest that:

(a) The normal reversal of OHC1-2 to adopt a lateral orientation needs to be maintained, at least transiently, and that maintenance also relies on GNAI/O (Results starting line 529. Disussion line 621).

(b) ptxA is more severe than Gnai2; Gnai3 when it comes to OHC1-2 orientation (Figure 9, role b). Oppositely, Gnai2; Gnai3 is obviously more severe when it comes to symmetry-breaking (Fig. 9, role a) and hair bundle morphogenesis (Fig. 9, c). It follows that the two early GNAI/O activities are qualitatively different and not just based on dose. This is essentially what this Reviewer correctly pointed out, and we have fully edited both Results and Discussion accordingly. We now speculate that the difference may lie in the identity of the necessary GNAI/O protein for each role. Any GNAI/O proteins acting as a switch downstream of the GPR156 receptor may relay orientation information (Fig. 9, role b), making ptxA a particularly effective disruption strategy since it downregulates all GNAI/O proteins. In contrast, symmetry-breaking may rely more specifically on GNAI2 and GNAI3, and ptxA is not expected to achieve a loss-of-function of GNAI2 and GNAI3 as extensive as a double targeted genetic inactivation of the corresponding genes. Please see new Results starting line 526 and Discussion starting line 603. We consequently abandoned the notion that increased doses of GNAI/O is required for each role, and we also clarify that symmetry-breaking (a) and orientation (b) occur at the same time (Fig. 9).

(2) P0 may not be late enough a stage to access phenotype maturity in the double mutants. For example, it is not clear from the basal PO results whether the IHC will acquire an inverted phenotype or just misorientation in the lateral side.

For context, the OHC1-2 misorientation pattern in the ptxA model at P0 does represent the end stage, as the same pattern is observed in adults (illustrated in Fig. 2A). In addition, OHC1-2 that express ptxA are inverted as soon as they break planar symmetry, and this was established at E16.5 in a previous publication where ptxA and Gpr156 misorientation patterns were compared and shown to be identical (Kindt et al., 2021 Supp. fig. 5C-D). However, we clearly failed to mention these important results in the original manuscript. We now cite Figure 2 for adult defects (line 522), and provide a citation for OHC1-2 inversion being observed from earliest stage of hair cell differentiation (Kindt et al., 2021) (line 519).

The vast majority of Gnai2; Gnai3 double mutants die before weaning but the single specimen we managed to collect at P21 also showed inverted OHC1-2 (representative example in Fig. 2A). Again, we previously failed to point out this important result. We now do so line 214 and 555. This is another evidence that OHC1-2 misorientation is in fact similar in the ptxA and Gnai2; Gnai3 models (but milder and delayed in the latter).

When it comes to IHCs and OHC3s however, the situation is less clear. These cell types are mildly misoriented in ptxA and Gpr156 mutants, but IHCs in particular appear severely misoriented in Gnai2; Gnai3 mutants based on the position of the basal body (Fig. 8). However, very dysmorphic hair bundles can pull on the basal body via the kinocilium and affect its position, which obscures hair cell orientation inferred from the basal body and subsequent interpretations. We do not delve on IHC and OHC3 and their orientation in Gnai2; Gnai3 mutants in the revision since we do not observe similar orientation defects in a different mouse model and lack sufficient adult data.

Suggestions to improve upon the manuscript for readers:(1) Line 294, indicate on the figure the staining in bare zone and tips of stereocilia on row 1.

Pertains to Figure 4. In A, we now point out the bare zone and stereocilia tips with arrow and arrowheads, respectively (as in other figures).

(2) Fig.8 schematic diagram, the labels of the line and 90o side by side is misleading.

We added black ticks for 0, 90, 180, 270 degree references. In contrast, the hair cell angle represented was switched to magenta.

(3) Fig. 7 legend, redundancy towards the end of the paragraph.

Thank you for catching this issue. A large portion of the legend was indeed accidentally repeated and is now deleted.

(4) Line 490-493, Another plausible explanation is that other factors besides Gnai2 and Gnai3 are involved in breaking symmetry during bundle establishment.

We now acknowledge that other proteins besides GNAI/O may be involved (Discussion line 614). That said, the notion that we do not achieve sufficient and/or early enough GNAI loss is supported for example by the Beer-Hammer 2018 study where no defects in symmetry-breaking or orientation were reported in their Gnai2 flox/flox; Gnai3 flox/flox model (Discussion new Line 637).

(5) Line 518, the base were largely inverted (Figure 8B). Should Fig 8A be cited instead of 8B?

Fig. 8B has graphs for the E17.5 cochlear base where OHC1-2 are inverted in both ptxA and Gnai2;3 DKO models. Fig. 8A has graphs of the E17.5 cochlear mid (less differentiated hair cells) where an inversion was not obvious previously, but is now clear although only partial in Gnai2; Gnai3 DKO (see above; raised eccentricity threshold). In the context of the previous text, this citation was thus correct. However, this section has been heavily modified to better compare Gnai2; Gnai3 DKO and ptxA and is hopefully less confusing in the revised version.

**Reviewer #2 (Public Review):**
Jarysta and colleagues set out to define how similar GNAI/O family members contribute to the shape and orientation of stereocilia bundles on auditory hair cells. Previous work demonstrated that loss of particular GNAI proteins, or inhibition of GNAIs by pertussis toxin, caused several defects in hair bundle morphogenesis, but open questions remained which the authors sought to address. Some of these questions include whether all phenotypes resulting from expression of pertussis toxin stemmed from GNAI inhibition; which GNAI family members are most critical for directing bundle development; whether GNAI proteins are needed for basal body movements that contribute to bundle patterning. These questions are important for understanding how tissue is patterned in response to planar cell polarity cues.To address questions related to the GNAI family in auditory hair cell development, the authors assembled an impressive and nearly comprehensive collection of mouse models. This approach allowed for each Gnai and Gnao gene to be knocked out individually or in combination with each other. Notably, a new floxed allele was generated for Gnai3 because loss of this gene in combination with Gnai2 deletion was known to be embryonic lethal. Besides these lines, a new knockin mouse was made to conditionally express untagged pertussis toxin following cre induction from a strong promoter. The breadth and complexity involved in generating and collecting these strains makes this study unique, and likely the authoritative last word on which GNAI proteins are needed for which aspect of auditory hair bundle development.Appropriate methods were employed by the authors to characterize auditory hair bundle morphology in each mouse line. Conclusions were carefully drawn from the data and largely based on excellent quantitative analysis. The main conclusions are that GNAI3 has the largest effect on hair bundle development. GNAI2 can compensate for GNAI3 loss in early development but incompletely in late development. The Gnai2 Gnai3 double mutant recapitulates nearly all the phenotypic effects associated with pertussis toxin expression and also reveals a role for GNAIs in early movement of the basal body. Although these results are not entirely unexpected based on earlier reports, the current results both uncover new functions and put putative functions on more solid ground.Based on this study, loss of GNAI1 and GNAO show a slight shortening of the tallest row of stereocilia but no other significant changes to bundle shape. Antibody staining shows no change in GNAI localization in the Gnai1 knockout, suggesting that little to no protein is found in hair cells. One caveat to this interpretation is that the antibody, while proposed to cross-react with GNAI1, is not clearly shown to immunolabel GNAI1. More than anything, this reservation mostly serves to illustrate how challenging it is to nail down every last detail. In turn, the comprehensive nature of the current study seems all the more impressive.

(1) The original manuscript quantified stereocilia properties in Gnai1 and Gnai2 single mutants, and in Gnai1; Gnai2 double mutants using non-parametric t-tests (Mann-Whitney) for comparisons. This approach indeed suggested subtle reduction in row 1 height in IHCs in all 3 mutants. We did not quantify stereocilia features in Gnao1 mutants but could not observe defects (new Fig. 2 Supp. 1E-F). In fact, we could not observe defects in Gnai1 and Gnai2 single mutants, and in Gnai1; Gnai2 double mutants either. For this reason we have been ambivalent about reporting defects for Gnai1 and Gnai2 single and Gnai1; Gnai2 double mutants.

In the revision, we applied a nested (hierarchical) t-test to avoid pseudo-replication (Eisner 2021; PMID: 33464305; https://pubmed.ncbi.nlm.nih.gov/33464305/). In our data, the nested t-tests structure measurements by animal instead of having all stereocilia or other cell measurements treated as independent values. This more stringent approach no longer finds row 1 height reduction significant in single Gnai1 or Gnai2 mutants, or in Gnai1; Gnai2 double mutants. We modified the text accordingly in Results and Discussion. Nested t-tests were applied uniformly across the manuscript and, besides IHC measurements in Fig. 2, now also apply to bare zone surface area in Fig. 6 and eccentricity in Fig. 7. For these experiments in contrast, previous conclusions are not changed. We think that this more careful statistical treatment is a closer representation of the data in term of the conclusions we can safely make.

(2) The reviewer's criticism about antibody specificity is accurate and fair, and is fully addressed in the revised manuscript. First, we provide a phylogeny cartoon as Figure 1A to compare the GNAI/O proteins and highlight how closely related they are in sequence. To validate the assumption that our approach would detect GNAI1 if it were present in hair cells, we took a new dual experimental approach in the revision. First, we electroporated Gnai1, Gnai2 and Gnai3 expression constructs in the E13.5 inner ear and tested whether the two GNAI antibodies used in the study can detect ectopic GNAI1 in Kolliker organ. This revealed that “ptGNAI2” detects GNAI1 very well (in addition to GNAI2), but that “scbtGNAI3” does not detect GNAI1 efficiently (although it does detect GNAI3 very well). To verify in vivo that “ptGNAI2” can detect endogenous GNAI1, we immunolabeled the gallbladder epithelium in Gnai1 mutants and littermate controls using the “ptGNAI2” antibody. Based on IMPC consortium data* about the Gnai1 LacZ mouse strain, Gnai1 is specifically expressed in the adult gallbladder. We could verify that signals detected in the Gnai1 mutants were visually reduced in comparison to littermate controls. We now added this validation step in Results line 309 and the data in Fig. 4 Supp. 1A-B.

*https://www.mousephenotype.org/data/genes/MGI:95771

**Reviewer #2 (Recommendations For The Authors):**
Minor comments that may marginally improve clarity.Abstract line 24: delete "nor polarized" because polarization cannot be assessed since the protein is undetectable.

This is a fair point, now deleted.

Consider revising: Lines 80-82; 188-202 (the order in which the mutants were presented was hard to follow for me); 239-240.

Lines 80-82: Used to read as "Ptx recapitulates severe stereocilia stunting and immature-looking hair bundles observed when GPSM2 or both GNAI2 and GNAI3 are inactivated."

Line 88: Was now changed to "Ptx provokes immature-looking hair bundles with severely stunted stereocilia, mimicking defects in Gpsm2 mutants and Gnai2; Gnai3 double mutants".

Lines 188-202: This was the first paragraph describing adult stereocilia defects in the different Gnai/o mouse strains. We completely rewrote the entire section to reflect the order in which the strains appear in Figure 2, hopefully making the text easier to follow because it better matches panels in Fig. 2 . We also made several other modifications to streamline comparisons and better introduce the orientation defects that are later detailed at neonate stages.

Lines 239-240: Used to read "GNAI2 makes a clear contribution since stereocilia defects increase in severity when GNAI loss extends from GNAI3 to both GNAI2 and GNAI3".

Line 247: Was now changed for "GNAI2 makes a clear contribution since Gnai3neo stereocilia defects dramatically increase in severity when GNAI2 is absent as well in Gnai2; Gnai3 double mutants."

Line 164: hardwired is unclear. Conserved?

We modified this sentence as follows: Line 171: "We reasoned that apical HC development is probably highly constrained and less likely to be influenced by genetic heterogeneity compared to susceptibility to disease, for example."

Line 299: It is not clear why GNAI1 is a better target than GNAI3. This phrase is repeated in line 303, I suspect inadvertently. Is there evidence that this antibody detects GNAI1, perhaps in another tissue? Line 308: GNAI1 may also not be detected by this antibody.

Please see point 2 above. We removed these hypothetical statements entirely and we instead now experimentally show that one of the two commercial antibodies used can readily detect GNAI1 (yet does not detect signal in hair cells when GNAI2 and GNAI3 are absent in Fig. 4F).